# Amplitude modulations of cortical sensory responses in pulsatile evidence accumulation

Sue Ann Koay[1], Stephan Thiberge[2], Carlos D Brody[1,3]*, David W Tank[1,2]*

[1]Princeton Neuroscience Institute, Princeton University, Princeton, United States; [2]Bezos Center for Neural Circuit Dynamics, Princeton University, Princeton, United States; [3]Howard Hughes Medical Institute, Princeton University, Princeton, United States

**Abstract** How does the brain internally represent a sequence of sensory information that jointly drives a decision-making behavior? Studies of perceptual decision-making have often assumed that sensory cortices provide noisy but otherwise veridical sensory inputs to downstream processes that accumulate and drive decisions. However, sensory processing in even the earliest sensory cortices can be systematically modified by various external and internal contexts. We recorded from neuronal populations across posterior cortex as mice performed a navigational decision-making task based on accumulating randomly timed pulses of visual evidence. Even in V1, only a small fraction of active neurons had sensory-like responses time-locked to each pulse. Here, we focus on how these 'cue-locked' neurons exhibited a variety of amplitude modulations from sensory to cognitive, notably by choice and accumulated evidence. These task-related modulations affected a large fraction of cue-locked neurons across posterior cortex, suggesting that future models of behavior should account for such influences.

**\*For correspondence:**
brody@princeton.edu (CDB);
dwtank@princeton.edu (DWT)

**Competing interests:** The authors declare that no competing interests exist.

## Introduction

As sensory information about the world is often noisy and/or ambiguous, an evidence accumulation process for increasing signal-to-noise ratio is thought to be fundamental to perceptual decision-making. Neural circuits that perform this are incompletely known, but canonically hypothesized to involve multiple stages starting from the detection of momentary sensory signals, which are then accumulated through time and later categorized into an appropriate behavioral action (*Gold and Shadlen, 2007*; *Brody and Hanks, 2016*; *Caballero et al., 2018*). In this picture, the sensory detection stage has a predominantly feedforward role, that is, providing input to but not otherwise involved in accumulation and decision formation. However, another large body of literature has demonstrated that sensory processing in even the earliest sensory cortices can be modified by various external and internal contexts, including motor feedback, temporal statistics, learned associations, and attentional control (*Roelfsema and de Lange, 2016*; *Gilbert and Sigman, 2007*; *Kimura, 2012*; *Gavornik and Bear, 2014*; *Glickfeld and Olsen, 2017*; *Niell and Stryker, 2010*; *Saleem et al., 2013*; *Shuler and Bear, 2006*; *Fiser et al., 2016*; *Haefner et al., 2016*; *Lee and Mumford, 2003*; *Zhang et al., 2014*; *Saleem et al., 2018*; *Makino and Komiyama, 2015*; *Keller et al., 2012*; *Poort et al., 2015*; *Li et al., 2004*; *Stănişor et al., 2013*; *Petreanu et al., 2012*; *Romo et al., 2002*; *Luna et al., 2005*; *Nienborg et al., 2012*; *Yang et al., 2016*; *Britten et al., 1996*; *Froudarakis et al., 2019*; *Keller and Mrsic-Flogel, 2018*). For example, feedback-based gain control of sensory responses has been suggested as an important mechanism for enhancing behaviorally relevant signals, while suppressing irrelevant signals (*Manita et al., 2015*; *Hillyard et al., 1998*; *Harris and Thiele, 2011*; *Azim and Seki, 2019*; *Douglas and Martin, 2007*; *Ahissar and Kleinfeld, 2003*).

The above two ideas—evidence accumulation and context-specific modulations—make two different but both compelling points about how sensory signals should be processed to support behavior. The two ideas are not mutually incompatible, and insight into the brain's specific implementation may be gained from a systematic investigation of sensory representations in the brain. To observe how each sensory increment influences neural dynamics, we utilized a behavioral paradigm with precisely initiated timings of sensory inputs that should drive an evidence accumulation process (*Brunton et al., 2013*). Specifically, we recorded from posterior cortical areas during a navigational decision-making task (*Pinto et al., 2018*; *BRAIN CoGS Collaboration, 2017*) where as mice ran down the central corridor of a virtual T-maze, pulses of visual evidence ('cues') randomly appeared along both left and right sides of the corridor. To obtain rewards, mice should accumulate the numerosities of cues, then turn down the maze arm corresponding to the side with more cues. The well-separated and randomized timing of cues allowed us to clearly identify putative sensory responses that were time-locked to each pulse, whereas the seconds-long periods over which cues were delivered allowed us to observe the timecourse of neural responses throughout a gradually unfolding decision.

Across posterior cortices, the bulk of neural activity was sequentially active vs. time in the trial, in a manner that did not depend directly on the sensory cues, as we describe in detail in another article (*Koay et al., 2019*). Even in the primary (V1) and secondary visual areas, only 5–15% of neurons active during the task had responses that were time-locked to sensory cues ('cue-locked cells'). Still, it is known that remarkably small signals on the order of a few cortical neurons can influence behavior (*Doron and Brecht, 2015*; *Buchan and Rowland, 2018*; *Tanke et al., 2018*; *Lerman et al., 2019*; *Carrillo-Reid et al., 2019*; *Marshel et al., 2019*). Here, we focused on the cue-locked cells, as candidates for momentary sensory inputs that may drive an accumulation and decision-making process. The responses of these cells to cues were well-described by a single impulse response function per neuron, but with amplitudes that varied across the many cue presentations. The cue-response amplitudes of most cells varied systematically across time in the trial, as well as across trials depending on behavioral context, thus suggesting gain modulation effects potentially related to decision-making dynamics. Across posterior cortices and including as early as in V1, these variations in cue-response amplitudes contained information about multiple visual, motor, cognitive, and memory-related contextual variables. Notably, in all areas about 50% of cue-locked cells had response amplitudes that depended on the choice reported by the animal at the end of the trial, or depended on the value of the gradually accumulating evidence. Top-down feedback, potentially from non-sensory regions in which the choice is formed, has been proposed to explain choice-related effects in sensory responses (*Britten et al., 1996*; *Romo et al., 2003*; *Nienborg and Cumming, 2009*; *Yang et al., 2016*; *Bondy et al., 2018*; *Wimmer et al., 2015*; *Haefner et al., 2016*). The dependence on accumulating evidence that we observed supports the hypothesis that this feedback may originate from an accumulator that itself eventually drives choice.

In sum, the amplitude modulations of cue-locked responses in this report can be thought of as due to multiplicative effects (or equivalently, changes in gain) on the brain's internal representation of individual sensory pulses. These multiplicative effects were moreover not entirely random from one cue to the next, but rather depended on task-specific factors including those that the brain presumably keeps track of using internal neural dynamics, such as the accumulated evidence. We thus suggest that psychophysical studies of pulsatile-evidence accumulation may benefit from considering that even at the earliest, sensory input stage, neural variability can have a component that is correlated across responses to multiple cues in a trial, as opposed to the independent noise often assumed under lack of knowledge otherwise. Our findings in this article point to candidate neural bases for temporally correlated noise that has been deduced from behavioral data to limit perceptual accuracy, for example in odor discrimination tasks for rats where the subject was free to continue acquiring sensory samples (*Zariwala et al., 2013*).

## Results

We used cellular-resolution two-photon imaging to record from six posterior cortical regions of 11 mice trained in the Accumulating-Towers task (*Figure 1a–c*). These mice were from transgenic lines that express the calcium-sensitive fluorescent indicator GCaMP6f in cortical excitatory neurons (Materials and methods), and prior to behavioral training underwent surgical implantation of an

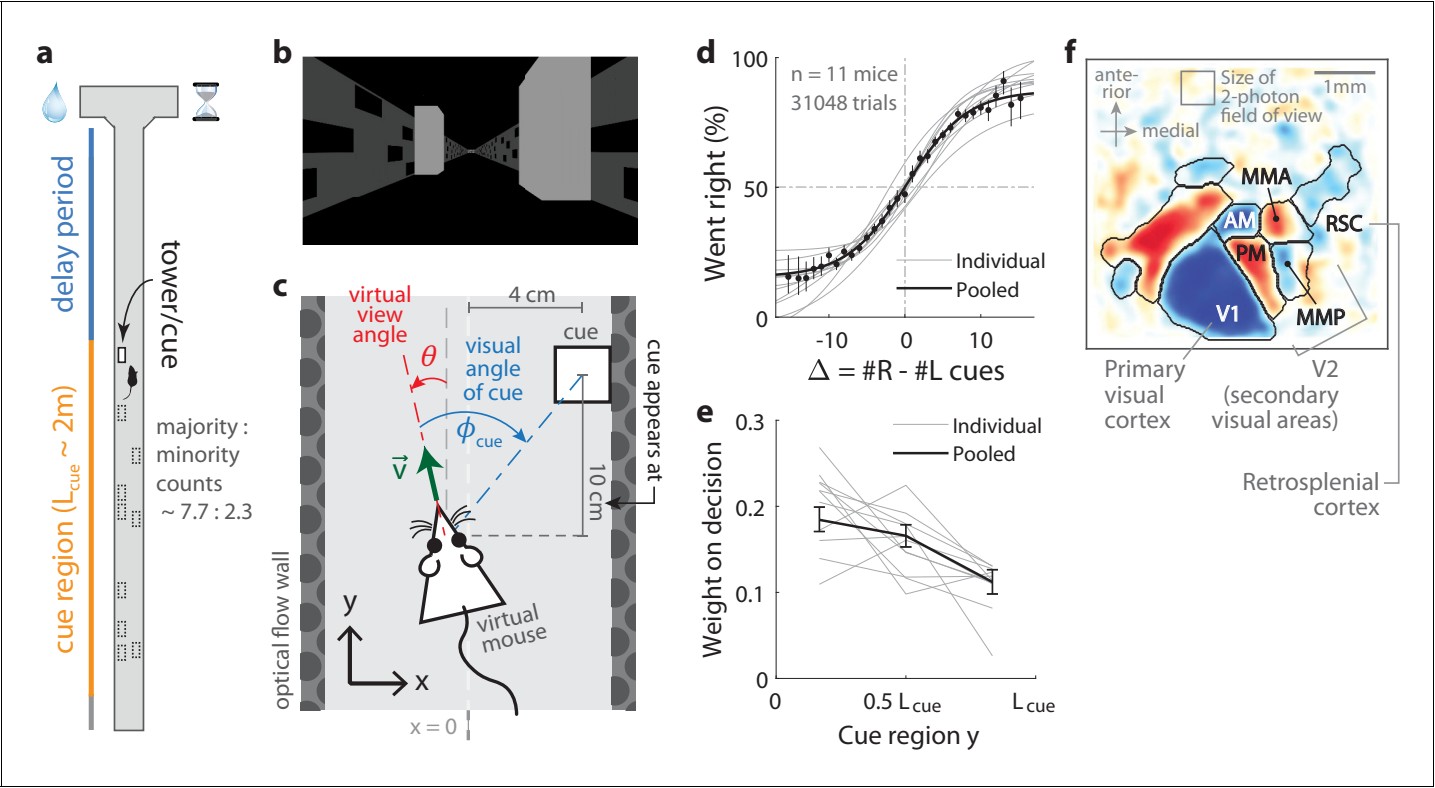

**Figure 1.** Two-photon calcium imaging of posterior cortical areas during a navigation-based evidence accumulation task. (**a**) Layout of the virtual T-maze in an example left-rewarded trial. (**b**) Example snapshot of the cue region corridor from a mouse's point of view when facing straight down the maze. Two cues on the right and left sides can be seen, closer and further from the mouse in that order. (**c**) Illustration of the virtual viewing angle θ. The visual angle $\phi_{cue}$ of a given cue is measured relative to θ and to the center of the cue. The $y$ spatial coordinate points straight down the stem of the maze, and the $x$ coordinate is transverse. $\vec{v}$ is the velocity of the mouse in the virtual world. (**d**) Sigmoid curve fits to behavioral data for how frequently mice turned right for a given difference in total right vs. total left cue counts at the end of the trial, $\Delta \equiv \#R - \#L$. Dots: Percent of trials (out of those with a given $\Delta$) in which mice turned right, pooling data from all mice. Error bars: 95% binomial C.I. (**e**) Logistic regression weights for predicting the mice's choice given spatially-binned evidence $\{\Delta_i\}$ where $i \in \{1, 2, 3\}$ indexes three equally sized spatial bins of the cue region. Error bars: 95% C.I. across bootstrap experiments. (**f**) Average visual field sign map ($n = 5$ mice) and visual area boundaries, with all recorded areas labeled. The visual field sign is $-1$ (dark blue) where the cortical layout is a mirror image and $+1$ (dark red) where it follows a non-inverted layout of the physical world. The online version of this article includes the following source data and figure supplement(s) for figure 1:

**Source data 1.** Data points, summary statistics, and kernel bandwidths.
**Figure supplement 1.** Session-specific behavioral parameters, by mouse.

optical cranial window centered over either the right or left parietal cortex. The mice then participated in previously detailed behavioral shaping (**Pinto et al., 2018**) and neural imaging procedures as summarized below.

Mice were trained in a head-fixed virtual reality system (**Dombeck et al., 2010**) to navigate in a T-maze. As they ran down the stem of the maze, a series of transient, randomly located tower-shaped cues (**Figure 1b,c**) appeared along the right and left walls of the cue region corridor (length $L_{cue} \approx 200\,cm$, average running speed in cue region $\approx 60\,cm/s$; see Materials and methods), followed by a delay region where no cues appeared. The locations of cues were drawn randomly per trial, with Poisson-distributed mean counts of 7.7 on the majority and 2.3 on the minority side, and mice were rewarded for turning down the arm corresponding to the side with more cues. In agreement with previous work (**Pinto et al., 2018**), all mice in this study exhibited characteristic psychometric curves (**Figure 1d**) and utilized multiple pieces of evidence to make decisions, with a small primacy effect (**Figure 1e**).

As the timing and visual location of the tower-shaped cues are important information about the behavior that we wished to relate to the neural activity, we programmed the virtual reality software to make a given cue visible to the mouse exactly when it reached a distance of 10 cm before the

cue's location along the T-maze stem (i.e. in the y coordinate, see *Figure 1c*). We refer to this instant at which a cue becomes visible as the 'onset time' for that cue, and used it to define the behavioral timings of visual pulses in all neural data analyses. Cues were made to vanish from view after 200 ms, although 1/11 mice ran so quickly that cues occasionally fell outside of the display range of the virtual reality system before that period (cue duration was ~190 ms for that mouse, see Materials and methods and *Figure 1—figure supplement 1* for details on timing precision). Lastly, as mice controlled the virtual viewing angle θ, cues could appear at a variety of visual angles $\phi_{cue}$ (*Figure 1c*). We accounted for this in all relevant data analyses, as well as conducted control experiments in which θ was restricted to be exactly zero from the beginning of the trial up to midway in the delay period (referred to as θ-controlled experiments; see Materials and methods).

For each mouse, we first identified the locations of the visual areas (*Figure 1f*; Materials and methods) using one-photon widefield imaging and a retinotopic visual stimulation protocol (*Zhuang et al., 2017*). Then, while the mice performed the task, we used two-photon imaging to record from $500 \mu m \times 500 \mu m$ fields of view in either layers 2/3 or 5 from one of six areas (*Supplementary file 1*, *Supplementary file 2*): the primary visual cortex (V1), secondary visual areas (V2 including anteromedial area [AM], posteromedial area [PM], medial-to-AM area [MMA], medial-to-PM area [MMP] [*Zhuang et al., 2017*]), and retrosplenial cortex (RSC). These fields of view were selected only to have good imaging quality (high apparent density of cells as unobscured as possible by brain vasculature), that is, prior to the start of the behavioral session and without any criteria based on neural responses. After correction for rigid brain motion, regions of interest representing putative single neurons were extracted using a semi-customized (Materials and methods) demixing and deconvolution procedure (*Pnevmatikakis et al., 2016*). The fluorescence-to-baseline ratio $\Delta F/F$ was used as an estimator of neural activity, and only cells with $\geq 0.1$ transients per trial were selected for analysis. In total, we analyzed 10,113 cells from 143 imaging sessions, focusing on 891 neurons identified as time-locked to the visual cues as explained in the next sections.

## Pulses of evidence evoke transient, time-locked responses in all recorded areas

We found neurons in all areas/layers that had activities clearly time-locked to the pulsatile cues (examples in *Figure 2a–b*). In trials with sparse occurrences of preferred-side cues, the activities of these cells tended to return to baseline following a fairly stereotyped impulse response. Individually, they thus represented only momentary information about the visual cues, although as a population they can form a more persistent stimulus memory (*Goldman, 2009*; *Scott et al., 2017*; *Miri et al., 2011*). Interestingly, the amplitudes of these cells' responses seemed to vary in a structured way, both across time in a trial, as well as across trials where the mouse eventually makes the choice to turn right vs. left (columns of *Figure 2a–b*). We therefore wished to quantify whether or not these putatively sensory amplitude changes also encoded other task-related information.

For a given cell, we estimated the amplitude of its response to each cue $i$ by modeling the cell's activity as a time series of non-negative amplitudes $A_i$ convolved with an impulse response function (*Figure 2c*). The latter was defined by lag, rise-time and fall-time parameters that were fit to each cell, but were the same for all cue-responses of that cell (deconvolving calcium dynamics; see Materials and methods). For a subset of neurons, this impulse response model resulted in excellent fits when the model included only primary responses to either right- or left-side cues (e.g. *Figure 2d*). In much rarer instances, adding a secondary response to the opposite-side cues resulted in a significantly better fit (e.g. *Figure 2e*; discounting for number of parameters by using AIC$_C$[*Hurvich and Tsai, 1989*] as a measure of goodness of fit). We defined cells to be cue-locked if the primary-response model yielded a much better fit to the data than a permutation test (data with cue timings shuffled within the cue region, see Materials and methods). For these cells, the trial-averaged activity predicted by the impulse response model (*Figure 2f*, magenta) was substantially larger than the magnitude of residuals of the fits (*Figure 2f*, 'data - model' prediction in black). For example, if cells had systematic rises or falls in baseline activity levels vs. time/place that could not be explained as transient responses to cues, then the residual would grow/diminish vs. y location in the cue region. *Figure 2—figure supplement 1a* shows that systematic trends (i.e. slopes) for the residual vs. y was small for most cells (68% C.I. of slopes across cells were within $[-0.098, 0.062]$, where a slope of $\pm 1$ corresponds to a change in residuals from the start to the end of the cue region being equal to the average signal predicted by the impulse response model). There were thus no large, unaccounted-

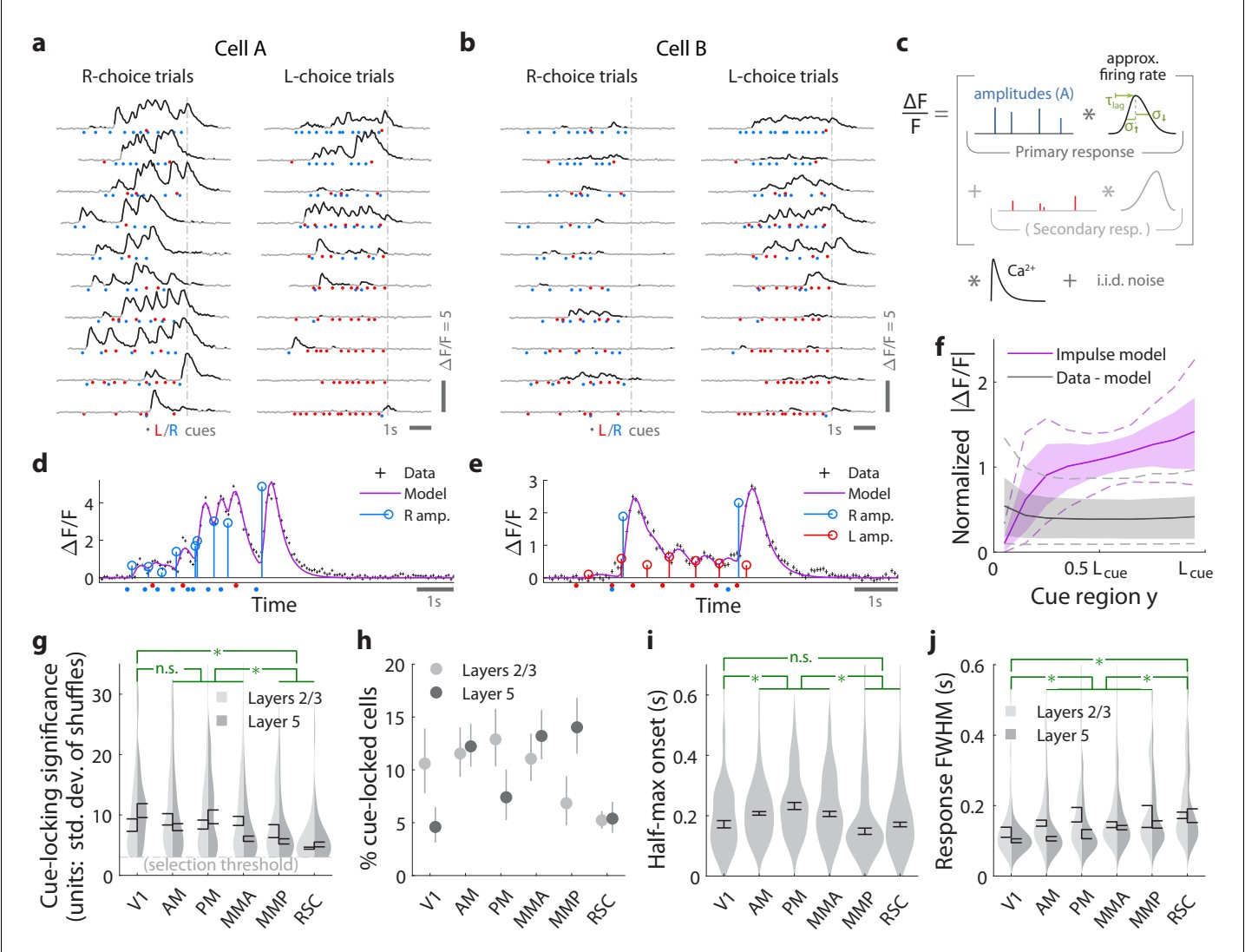

**Figure 2.** Pulses of evidence evoke transient, time-locked responses that are well described by an impulse response model. (**a**) Trial-by-trial activity (rows) vs. time of an example right-cue-locked cell recorded in area AM, aligned in time to the end of the cue period (dashed line). Onset times of left (right) cues in each trial are shown as red (blue) dots. (**b**) Same as (**a**), but for an atypical right-cue-locked cell (in area AM) that has some left-cue-locked responses. (**c**) Depiction of the impulse response model for the activity level $\Delta F/F$ of a neuron vs. time (x-axis). Star indicates the convolution operator. (**d**) Prediction of the impulse response model for the cell in (**a**) in one example trial. This cell had no significant secondary (left-cue) responses. (**e**) Same as (**d**) but for the cell in (**b**). The model prediction is the sum of primary (right-cue) and secondary (left-cue) responses. (**f**) Trial-average impulse response model prediction (purple) vs. the residual of the fit ($\Delta F/F$ data minus model prediction, black), in 10 equally sized spatial bins of the cue region. For a given cell, the average model prediction (or average residual) is computed in each spatial bin, then the absolute value of this quantity is averaged across trials, separately per spatial bin. Line: Mean across cells. Dashed line: 95% C.I. across cells. Band: 68% C.I. across cells. For comparability across cells, $\Delta F/F$ was expressed in units such that the mean model prediction of each cell is 1. The model prediction rises gradually from baseline at the beginning of the cue period due to nonzero lags in response onsets. (**g**) Distribution (kernel density estimate) of cue-locking significance for cells in various areas/layers. Significance is defined per cell, as the number of standard deviations beyond the median $AIC_C$ score of models constructed using shuffled data (Materials and methods). Error bars: S.E.M. of cells. Stars: significant differences in means (Wilcoxon rank-sum test). (**h**) Percent of significantly cue-locked cells in various areas/layers. Chance: $10^{-3}$%. Error bars: 95% binomial C.I. across sessions. (**i**) Distribution (kernel density estimate) of the half-maximum onset time of the primary response, for cells in various areas. Data were pooled across layers (inter-layer differences not significant). Error bars: S.E.M. across cells. Stars: significant differences in means (Wilcoxon rank-sum test). (**j**) As in (**i**) but for the full-width-at-half-max. Statistical tests use data pooled across layers. Means were significantly different across layers for areas AM and PM (Wilcoxon rank-sum test). The online version of this article includes the following source data and figure supplement(s) for figure 2:

**Source data 1.** Data points including individual entries for histograms, summary statistics and kernel bandwidths.

**Figure supplement 1.** Additional statistics for cue-locked responses.

for components in the activity of these identified cue-locked cells, in particular no components with long timescales.

Significantly cue-locked cells comprised a small fraction of the overall neural activity, but were nevertheless present in all areas/layers and exhibited some progression of response properties across posterior cortical areas in a roughly lateral-to-medial order (V1, V2, RSC). Cells with the most precisely time-locked responses to cues were found in the visual areas as opposed to RSC (high-significance tail of distributions in *Figure 2g*; low significance means that the model fit comparably well to data where cue timings were shuffled within the cue region). Reflecting this, about 5–15% of cells in visual areas were significantly cue-locked, compared to ~5% in RSC (*Figure 2h*). Of these significant cells, only ~3% had secondary responses that were moreover much less significantly time-locked (*Figure 2—figure supplement 1b*); most cells responded to only contralateral cues (*Figure 2—figure supplement 1c*). The onset of the half-maximum response was ~200 ms after each pulse (*Figure 2i*), and the response full-width-at-half-max (FWHM) was ~100 ms but increased from V1 to secondary visual areas to RSC (*Figure 2j*). The impulse response model thus identified cells that follow what one might expect of purely visual-sensory responses on a cue-by-cue basis, but up to amplitude changes that we next discuss.

## Cue-locked response amplitudes contain information about visual, motor, cognitive, and memory-related contextual task variables

Studies of perceptual decision-making have shown that the animal's upcoming choice affects the activity of stimulus-selective neurons in a variety of areas (*Britten et al., 1996*; *Nienborg and Cumming, 2009*). We analogously looked for such effects (and more) while accounting for the highly dynamical nature of our task. As neurons responded predominantly to only one laterality of cues, all our subsequent analyses focus on the primary-response amplitudes of cue-locked cells. Importantly, the impulse response model deconvolves responses to individual cues, so the response amplitude $A_i$ can be conceptualized as a multiplicative gain factor that the cell's response was subject to at the instant at which the $i^{th}$ cue appeared.

We used a neural-population decoding analysis to quantify how much information the cue-locked response amplitudes contained about various contextual variables. First, for the $i^{th}$ cue in the trial, we defined the neural state as the vector of amplitudes $A_i$ of cells that responded to contralateral cues only. Then using the neural states corresponding to cues that occurred in the first third of the cue period, we trained a support vector machine (SVM) to linearly decode a given task variable from these neural states (cross-validated and corrected for multiple comparisons; see Materials and methods). This procedure was repeated for the other two spatial bins (second third and final third) of the cue period, to observe changes in neural information that may reflect place-/time-related changes in task conditions (illustrated in *Figure 3a*). *Figure 3b* shows that across posterior cortex, four task variables were accurately decodable from the cue-response amplitudes: the view angle $\theta$, running speed, the running tally of evidence ($\Delta \equiv \#R - \#L$), and the eventual choice to turn right or left. The reward outcome from the previous trial could also be decoded, albeit less accurately, while in contrast decoding of the past-trial choice was near chance levels.

As the six variables were statistically correlated by nature of the task (e.g. the mouse controls $\theta$ to execute the navigational choice, *Figure 3a*), indirect neural information about one variable could be exploited to increase the performance of decoding another correlated variable (*Krumin et al., 2018*; *Koay et al., 2019*). To account for this, we repeated the decoding analyses for a modified set of variables that had statistical correlations removed. As explained in the Materials and methods and illustrated in *Figure 3c*, we solved for uncorrelated modes being linear combinations of the original time-traces that were closest, in the least-squares sense, to the original traces, while constrained to be themselves uncorrelated with each other. As inter-variable correlations were low throughout the cue region, these uncorrelated modes were very similar to the original task variables. Each uncorrelated mode was identified with its closest original variable and labeled as such, and correlation coefficients between individual uncorrelated modes and their corresponding original variables were > 0.85 for all modes (*Figure 3—figure supplement 1*). Performances for decoding the uncorrelated modes were a little lower than the original task variables (*Figure 3d*), as expected since contributions from indirect neural information could no longer be present. Nevertheless, the modes that resembled view angle, speed, evidence, choice, and past-trial reward could all be consistently

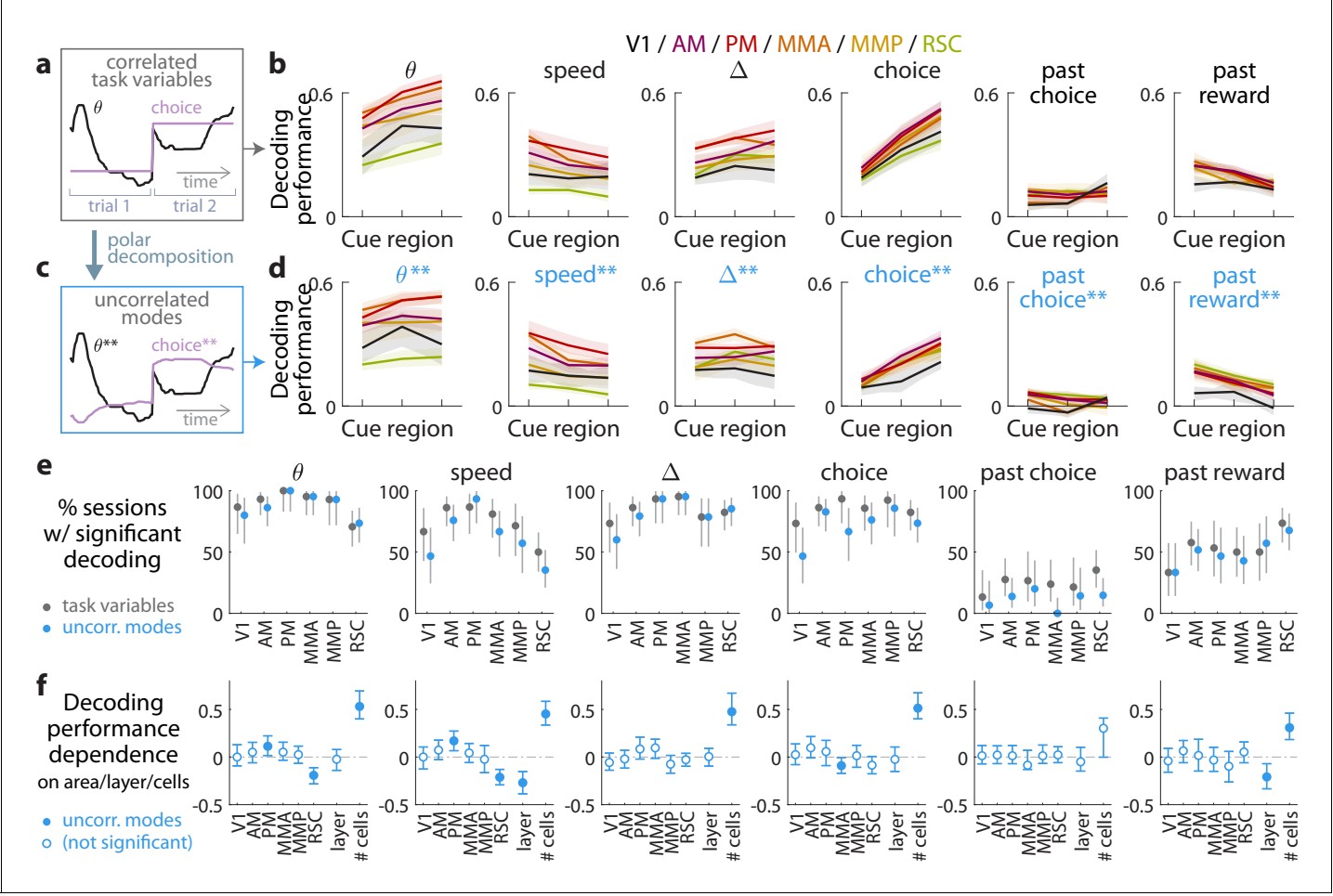

**Figure 3.** Multiple visual, motor, cognitive, and memory-related variables can be decoded from the amplitudes of cue-locked cell responses. (a) Example time-traces of two statistically correlated task variables, the view angle θ (black) and the eventual navigational choice (magenta). (b) Cross-validated performance for decoding six task variables (individual plots) from the amplitudes of cue-locked neuronal responses, separately evaluated using responses to cues in three spatial bins of the cue region (Materials and methods). The performance measure is Pearson's correlation between the actual task variable value and the prediction using cue-locked cell amplitudes. Lines: mean performance across recording sessions for various areas (colors). Bands: S.E.M. across sessions, for each area. (c) Example time-traces of the two uncorrelated modes obtained from a polar decomposition of the correlated task variables in (a). This decomposition (Materials and methods) solves for these uncorrelated modes such that they were linear combinations of the original time-traces that were closest, in the least-squares sense, to the original traces, while constrained to be themselves uncorrelated with each other. Correlation coefficients between individual uncorrelated modes and their corresponding original variables were > 0.85 for all modes (*Figure 3—figure supplement 1*). (d) As in (a), but for decoding the uncorrelated task-variable modes illustrated in (c). (e) Proportion of imaging sessions that had significant decoding performance for the six task variables in (b) (dark gray points) and uncorrelated modes in (d) (blue points), compared to shuffled data and corrected for multiple comparisons. Data were restricted to 140/143 sessions with at least one cue-locked cell. Error bars: 95% binomial C.I. across sessions. (f) Linear regression (Support Vector Machine) weights for how much the decoding performance for uncorrelated task-variable modes in (d) depended on cortical area/layer and number of recorded cue-locked cells. The decoder accuracy was evaluated at the middle of the cue region for each dataset. The area and layer regressors are indicator variables, e.g. a recording from layer 5 of V1 would have regressor values (V1 = 1, AM = 0, PM = 0, MMA = 0, MMP = 0, RSC = 0, layer = 1). Weights that are not statistically different from zero are indicated with open circles. The negative weight for layer dependence of past-reward decoding means that layer five had significantly lower decoding performance than layers 2/3. Error bars: 95% C.I. computed via bootstrapping sessions.

The online version of this article includes the following source data and figure supplement(s) for figure 3:

**Source data 1.** Data points and summary statistics.

**Figure supplement 1.** Pearson's correlation between uncorrelated behavioral modes (θ**, speed**, etc.) and the corresponding most similar task variable (θ, speed, etc.).

**Figure supplement 2.** Qualitatively similar performances for decoding task variables from cue-locked response amplitudes in control experiments with view-angle restricted to zero in the cue region.

**Figure supplement 3.** Evidence (and other task variables) can still be decoded from cue-locked response amplitudes, excluding cells that exhibit stimulus-specific adaptation (SSA).

decoded across imaging sessions for all examined areas (*Figure 3e*). There was also comparably high performance of decoding evidence and choice in the θ-controlled experiments (*Figure 3—figure supplement 2*), which explicitly shows that neural information about these variables do not originate solely from changes in visual perspective. In a comparable task where choice was highly correlated with view angle (θ) and *y* spatial location in the maze, it has previously been reported that θ and *y* explains most of neural responses in parietal posterior cortex, with small gains from including choice as a third factor (*Krumin et al., 2018*). Interestingly however, our findings indicate that in a task where choice was *distinguishable* from other behavioral factors (here, at least within the cue region), there was significant neural information in all examined posterior cortical areas about this internally generated variable, choice.

As a population, the amplitudes of cue-locked cells thus reflected a rich set of present- and past-trial contextual information, with some apparent anatomical differences seen in *Figure 3b,d*. However, instead of the neural representation being different across different cortical regions, an alternative explanation could be that the accuracy of decoding task variable information depended on experimental factors such as the number of recorded neurons (which differed systematically across cortical areas/layers). To address this, we constructed a linear regression model to predict the decoding accuracy for various datasets as a weighted sum of a set of factors: the cortical area, layer, and number of recorded cells. The cortical area and layer regressors are indicator variables (0 or 1) that specify whether a given dataset was a recording from a particular area and layers 2/3 vs. 5. Likely due to the small numbers of recorded cue-locked cells per session (~0–10), the decoding performance for all variables depended most strongly on the number of cells (*Figure 3f*). *Figure 3f* also shows that RSC had significantly lower view angle and speed decoding performance than other regions, which we can think of as increased invariance of cue-locked response amplitudes to low-level visual parameters of the stimuli. Layer 5 was also distinguishable from layer 2/3 data in having reduced performance for decoding speed and past-trial reward.

## Decision-related changes in cue-locked response amplitudes are compatible with a feedback origin

Interestingly, the response amplitudes of some individual cue-locked cells appeared to systematically depend on time (e.g. *Figure 2a–b*), as did the population-level decoding performance for variables such as choice (*Figure 3b,d*). To understand if these neural dynamics may reflect a gradually unfolding decision-making process, we turned to modeling how amplitudes of cue-locked cell responses may depend on choice and place/time, while accounting for other time-varying behavioral factors.

As a null hypothesis based on previous literature, we hypothesized that cue-response amplitudes can depend on a receptive field specified by the visual angle of the cue ($\phi_{cue}$, *Figure 1c*), as well as running speed (*Niell and Stryker, 2010*; *Saleem et al., 2013*). Given limited data statistics, we compared this null hypothesis to three other conceptually distinct models (Materials and methods), each of which aims to parsimoniously explain cue-response amplitudes using small sets of behavioral factors. These models predict the observed cue-response amplitudes to be random samples from a Gamma distribution, where the mean of the Gamma distribution is a function of various behavioral factors at the time at which a given cue appeared. The mean functions for all models have the form $\rho(\phi_{cue}) \, f(v) \, g(\cdots)$, where $\rho(\phi_{cue})$ is an angular receptive field function, $f(v)$ is a running speed ($v$) dependence function, and $g(\cdots)$ is specific to each of the three models, as follows. First, the 'SSA' model parameterizes stimulus-specific adaptation (*Ulanovsky et al., 2003*; *Sobotka and Ringo, 1994*) or enhancement (*Vinken et al., 2017*; *Kaneko et al., 2017*) with exponential time-recovery in between cues. Second, the 'choice' model allows for a flexible change in amplitudes vs. place/time in the cue region, with a potentially different trend for right- vs. left-choice trials. Third, the 'cue-counts' model allows the amplitudes to depend on the running tally of $\#R$, $\#L$, or $\Delta = \#R - \#L$. This selection of models allows us to ask if cue-locked responses are sufficiently explained by previously known effects, or if after accounting for such there are still effects related to the accumulation process, such as choice or cue-count dependence.

We constructed the amplitude model prediction as the $AIC_C$-likelihood-weighted average of the above models, which accounts for when two or more are comparably good (*Volinsky et al., 1999*). As illustrative examples, *Figure 4a* shows how the amplitudes of two simultaneously recorded cue-locked cells in area AM depended on behavioral factors and compared to model predictions. There are clear differences in predictions for right- vs. left-choice trials that can also be seen in the raw

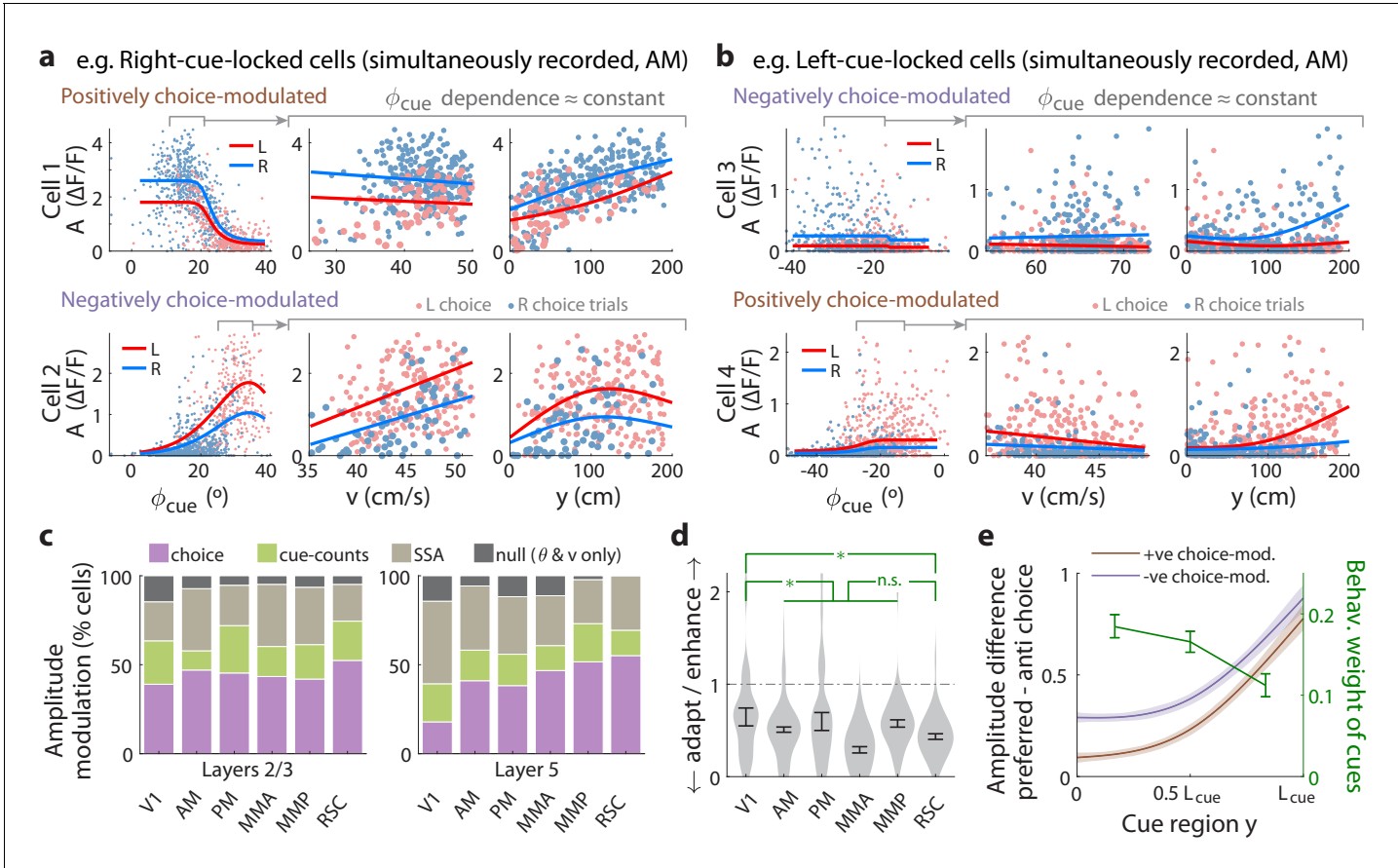

**Figure 4.** Cue-locked response amplitudes depend on view angle, speed, and cue frequency, but a large fraction exhibit choice-related modulations that increase during the course of the trial. (a) Response amplitudes of two example right-cue-locked cells (one cell per row) vs. (columns) the visual angle at which the cue appeared ($\phi_{cue}$), running speed (*v*), and *y* location of the cue in the cue region. Points: amplitude data in blue (red) according to the upcoming right (left) choice. Lines: AIC$_C$-weighted model mean functions for right- vs. left-choice trials (lines); the model predicts the data to be random samples from a Gamma distribution with this behavior-dependent mean function. The data in the right two columns were restricted to a subset where angular receptive field effects are small, corresponding to the indicated area in the leftmost plots. (b) Same as (a) but for two (left-cue-locked) cells with broader angular receptive fields. (c) Percentages of cells that significantly favor various amplitude modulation models (likelihood ratio <0.05, defaulting to null model if none are significant), in the indicated cortical areas and layers. For layer 2/3 data, V1 has a significantly higher fraction of cells preferring the null model than other areas (p = 0.02, two-tailed Wilcoxon rank-sum test). For layer 5 data, V1 has a significantly lower choice-model preferring fraction than the other areas (p = 0.003). (d) Distribution (kernel density estimate) of adaptation/enhancement factors for cells that favor the SSA model. A factor of 1 corresponds to no adaptation, while for other values the subsequent response is scaled by this amount with exponential recovery toward 1. Error bars: S.E.M. Stars: significant differences in means (Wilcoxon rank-sum test). (e) Comparison of the behaviorally deduced weighting of cues (green, same as *Figure 1e*) to the neural choice modulation strength vs. location in the cue region (for contralateral-cue-locked cells only, but ipsilateral-cue-locked cells in *Figure 4—figure supplement 2g* have similar trends). The choice modulation strength is defined using the amplitude-modulation model predictions, and is the difference between predicted amplitudes on preferred-choice minus anti-preferred-choice trials, where preferred choice means that the neuron will have higher amplitudes on trials of that choice compared to trials of the opposite (anti-preferred) choice. For comparability across cells, the choice modulation strength is normalized to the average amplitude for each cell (Materials and methods). Lines: mean across cue-locked cells, computed separately for positively vs. negatively choice-modulated cells (data from all brain regions). Bands: S.E.M.

The online version of this article includes the following source data and figure supplement(s) for figure 4:

**Source data 1.** Data points and summary statistics.

**Figure supplement 1.** Qualitatively similar cue-locked amplitude modulations in control experiments with view angle restricted to be zero in the cue region.

**Figure supplement 2.** Additional statistics for amplitude modulations of cue-locked cells.

**Figure supplement 2—source data 1.** Summary statistics.

amplitude data (restricted to a range of $\phi_{cue}$ such that angular receptive field effects are small, 2nd and 3rd columns of *Figure 4a*). Although both cells responded preferentially to right-side cues, they had oppositely signed choice modulation effects, defined as the difference between amplitude model predictions on contralateral- vs. ipsilateral-choice trials (Materials and methods). *Figure 4b* shows two more example choice-modulated cells that had near-constant angular receptive fields. We note that except for the parameters of SSA, all findings in this section were qualitatively similar in θ-controlled experiments where there can be no angular receptive field effects (*Figure 4—figure supplement 1*).

To summarize the prevalence and composition of amplitude-modulation effects, we selected the best model per cell using $AIC_C$, defaulting in ambiguous cases (relative likelihood <0.05) to the null hypothesis. *Figure 4—figure supplement 2a* shows that there were large fractions of cells with very high $AIC_C$ likelihoods for all three alternative models compared to the null hypothesis. Cells that favored the cue-counts model could also be clearly distinguished from those that favored the SSA model (*Figure 4—figure supplement 2b*); in fact, exclusion of cells that exhibited SSA had little effect on how well evidence and other variables could be decoded from the neural population (*Figure 3—figure supplement 3*). In all areas and layers, >85% of cue-locked cells exhibited some form of amplitude modulations beyond angular receptive field and running speed effects (*Figure 4c*). Overall, $27^{+4}_{-4}\%$ of cells were best explained by SSA while $67^{+4}_{-4}\%$ favored either choice or cue-counts models. The notable inter-area difference is for layer 5 data, which had a qualitatively smaller proportion of choice-model preferring cells in V1 compared to other areas (p = 0.003, Wilcoxon rank-sum test). Most cells thus exhibited some form of amplitude modulations beyond visuomotor effects, with little difference in composition across areas and layers.

Although SSA, choice, and cue-counts dependencies all predict changes in cue-response amplitudes vs. time in the trial, there were qualitative differences that distinguished SSA from choice and cue-count modulations, as we next discuss. Cells in the two largest categories, SSA and choice, had qualitatively different population statistics for how their cue-response amplitudes depended on place/time in the trial. Most cells ($92^{+3}_{-4}\%$) that favored the SSA model corresponded to a phenotype with decreased responses to subsequent cues. Adaptation effects were weakest in V1 and stronger other areas (*Figure 4d*, but see *Figure 4—figure supplement 1f–g* for θ-controlled experiments), although the ~0.8 s recovery timescale had no significant inter-area differences (*Figure 4—figure supplement 2d*). In contrast, cue-locked cells with both choice laterality preferences were intermixed in all areas and layers (*Figure 4—figure supplement 2e*). Also unlike the *decrease* in response amplitudes vs. time for cells that favored the SSA model, both subpopulations of positively and negatively choice-modulated cells exhibited gradually *increasing* effect sizes vs. place/time in the trial (*Figure 4e* for contralateral cue-locked cells, *Figure 4—figure supplement 2g* for ipsilateral cue-locked cells). Cells that favored the cue-counts modulated category also had qualitatively different population statistics compared to cells that exhibited SSA. Comparable proportions of cue-counts modulated cells were best explained by dependence on counts on either the contralateral side, ipsilateral side, or the difference of the two sides (*Figure 4—figure supplement 2f*). For (say) right-cue-locked cells, $\#L$ or $\Delta$ dependencies are not directly explainable by SSA because the modulation is by left-side cues that the cells do not otherwise respond to. The remaining time-independent $\#R$ modulation also cannot be explained by SSA, unless SSA has an infinitely long timescale. Such infinite-timescale SSA would require some additional prescription for 'resetting' the adaptation factor, for example at the start of each trial, because otherwise amplitudes would continue to decrease/increase throughout the ~1 hr long session (which we do not observe).

Although relationships between sensory responses and choice can arise in a purely feedforward circuit structure, because sensory neurons play a causal role in producing the behavioral choice (*Shadlen et al., 1996*), others have noted that this should result in similar timecourses of neural and behavioral fluctuations (*Nienborg and Cumming, 2009*). Instead, we observed contrasting timecourses: as each trial evolved, there was a slow increase in time in choice modulations of cue-locked responses (*Figure 4e*; *Figure 4—figure supplement 2g*), which was opposite to the behaviorally-assessed decrease in time in how sensory evidence fluctuations influenced the mice's choice (green line in *Figure 4e*, which was replicated from *Figure 1e*). Additionally, a feedforward structure predicts that positive fluctuations in right- (left-)preferring cue-locked neurons should produce rightwards (leftwards) fluctuations in choice. Instead, we observed that about half of the cue-locked cells

were modulated by choice in a manner opposite to their cue-side preference (*Figure 4—figure supplement 2e*). Both of these observations argue against a purely feedforward structure, and thus support the existence of feedback influences on sensory responses (*Wimmer et al., 2015*; *Nienborg and Cumming, 2009*; *Haefner et al., 2016*).

## Discussion

Psychophysics-motivated evidence accumulation models *Ratcliff and McKoon, 2008*; *Stone, 1960*; *Bogacz et al., 2006* have long guided research into how such algorithms may map onto neural activity and areas in the brain. A complementary, bottom-up approach starts from data-driven observations and formulates hypotheses based on the structure of the observations (*Shadlen et al., 1996*; *Wimmer et al., 2015*). In this direction, we exploited the mouse model system to systematically record from layers 2/3 and 5 of six posterior cortical areas during a task involving temporal accumulation of pulsatile visual evidence. A separate optogenetic perturbation study showed that all of these areas contributed to mice's performance of the Accumulating-Towers task (*Pinto et al., 2019*). We reasoned that to understand how cortical areas contribute to evidence accumulation, a necessary first step is to understand the neural representation of sensory inputs to the process. In this work, we therefore focused on cue-locked cells that had sensory-like responses that is, time-locked to individual pulses of evidence, which comprised ~5–15% of active neurons in visual areas and ~5% in the RSC. These cells are candidates for sensory inputs that may feed into an accumulation process that drives behavior, but could also reflect more complex neural dynamics such as from top-down feedback throughout the seconds-long decision formation process. We characterized properties of cue-locked responses across the posterior cortex, which revealed that although we selected cells that had highly stereotypical time-courses of impulse responses to individual cues, the amplitudes of these responses varied across cue presentations in intriguingly task-specific ways.

One long-standing postulated function of the visual cortical hierarchy is to generate invariant visual representations (*DiCarlo et al., 2012*), for example for the visual cues regardless of viewing perspective or placement in the T-maze. On the other hand, predictive processing theories propose that visual processing intricately incorporates multiple external and internal contextual information, in a continuous loop of hypothesis formation and checking (*Rao and Ballard, 1999*; *Bastos et al., 2012*; *Keller and Mrsic-Flogel, 2018*). Compatible with the latter hypotheses, we observed that across posterior cortices, cue-locked cells had amplitude modulations that reflected not only visual perspective and running speed (*Niell and Stryker, 2010*; *Saleem et al., 2013*), but also the accumulated evidence, choice, and reward history (neural population decoding in *Figure 3*). Inter-area differences were mostly in degree (*Minderer et al., 2019*), with V1 having significantly lower performance for decoding view angle and choice, whereas RSC had lower decoding performance for speed but higher decoding performance for evidence (*Figure 3f*). We also observed an anatomical progression from V1 to secondary visual areas to RSC in terms of increasing timescales of cue-locked responses (*Figure 2i–j*) and increasing strengths of stimulus-specific adaptation (*Figure 4d*). Our results are compatible with other experimental findings of increasing timescales along a cortical hierarchy (*Murray et al., 2014*; *Runyan et al., 2017*; *Dotson et al., 2018*; *Schmolesky et al., 1998*), and theoretical proposals that all cortical circuits contribute to accumulation with intrinsic timescales that follow a progression across brain areas (*Hasson et al., 2015*; *Chaudhuri et al., 2015*; *Christophel et al., 2017*; *Sreenivasan et al., 2014*).

The amplitude modulations of cue-locked cells can be interpreted as multiplicative gain changes on otherwise sensory responses, and could be clearly distinguished from additive effects due to our experimental design with pulsatile stimuli and high signal-to-noise calcium imaging (*Figure 2*). While a number of other studies have quantified the presence of multiplicative noise correlations in cortical responses (*Goris et al., 2014*; *Arandia-Romero et al., 2016*; *Lin et al., 2015*), we showed that for most cells the amplitude variations were not random, but instead depended systematically on visuomotor and cognitive variables (*Figure 4c*). Relationships between sensory responses and choice can arise in a purely feedforward circuit structure (*Shadlen et al., 1996*), where the causal role of sensory neurons in producing the behavioral choice predicts that choice-related neural and behavioral fluctuations should have similar timecourses (*Nienborg and Cumming, 2009*). Incompatible with a solely feedforward circuit hypothesis, we instead observed that choice modulations of cue-locked responses *increased* in time (*Figure 4e*), whereas the behavioral influence of sensory evidence

fluctuations on the mice's choice *decreased* in time (*Figure 1e*). Both the choice- and count-modulation observations discussed here were suggestive of signals originating from an accumulator.

Our findings extend previous reports of relationships between sensory responses and perceptual decisions, termed 'choice probability (*Britten et al., 1996*)' (CP), and may constitute a form of conjunctive coding of cue and contextual information that preserves both the specificity and precise timing of responses to cues. An interesting question arises as to whether such multiplexing of cue and contextual information can cause potential interference between the different multiplexed information. For example, many evidence accumulation studies have reported positive correlations between CP and the stimulus selectivity of cells (*Britten et al., 1996*; *Celebrini and Newsome, 1994*; *Cohen and Newsome, 2009*; *Dodd et al., 2001*; *Law and Gold, 2009*; *Price and Born, 2010*; *Kumano et al., 2016*; *Sasaki and Uka, 2009*; *Gu et al., 2014*; *Nienborg and Cumming, 2014*) (for a differing view, see *Zaidel et al., 2017* for analyses that better separate effects of stimulus vs. choice responses, and *Zhao et al., 2020* for a recent re-analysis at the neural-population level). Translated to our task, positively correlated CP vs. stimulus preferences means that neurons that responded selectively to *right* cues tended to have increased firing rates when the animal will make a choice to the *right*. In this kind of coding scheme, increased activity in right-cue-locked cells could be due to either more right-side cues being presented or an internally generated right-choice signal, and there is no obvious way to distinguish between these two possibilities from just the activities of these cells. Our data deviates from the abovementioned CP studies in that highly contralateral-cue-selective neurons could be divided into two near-equally sized subpopulations with positive choice modulation (analogous to CP >0.5) and negative choice modulation (CP <0.5) respectively (*Figure 4—figure supplement 2e*). As two simultaneously recorded cells that respond to the *same* visual cue can be *oppositely* modulated (*Figure 4a*), these phenomena are not expected from canonical accounts of spatial- or feature/object-based attention in visual processing (*Cohen and Maunsell, 2014*; *Treue, 2014*), but rather more compatible with mixed choice- and sensory-selectivity reported in other perceptual decision-making experiments (*Raposo et al., 2014*).

We can conceptualize how our CP-related findings differ from previous literature by considering how choice modifies the neural-population-level representations of the visual cues, as illustrated in *Figure 5* for two hypothetical neurons that both respond to right-side cues. We refer to the joint activity levels of these two hypothetical neurons as the neural (population) state. *Figure 5a* illustrates that when there is no cue both neurons have near-zero activity levels (gray dots), whereas when a right-side cue is present both neurons have high activity levels with some variations due to noise (purple dots). Conversely, the presence or absence of a right-side cue can be better decoded from the neural-population activity than from individual noisy neurons, by summing their activities or equivalently projecting the two-dimensional neural state onto a cue-decoding direction $\vec{d}_{cue}$ as depicted in *Figure 5a*. If in addition these two neurons both have CP >0.5 (positive choice modulation), this means that the neural responses in the presence of a right-side cue can further be separated into two distinguishable distributions depending on whether the subject will eventually make a right or left behavioral choice (*Figure 5b*, blue or red dots for the two choices, respectively). The CP >0.5 case corresponds to both neurons having slightly higher (lower) activity levels on right (left) choice trials, which means that we can decode the subject's behavioral choice by projecting the neural state onto a choice-decoding direction $\vec{d}_{choice}$ that is more or less aligned with the cue-decoding direction $\vec{d}_{cue}$ (arrows in *Figure 5b*). However as noted above, collinearity of $\vec{d}_{choice}$ with $\vec{d}_{cue}$ means that based on neural activities alone, there can be many cases where we cannot unambiguously decide whether the subject saw more right-side cues or will make a right behavioral choice (overlap between blue and red points in *Figure 5b*). This is distinct from the case—as observed in our data—where the two neurons have opposite choice modulations, for example neuron 1 has CP <0.5 (negative choice modulation) whereas neuron 2 has CP >0.5. As depicted in *Figure 5c*, neuron 1 now has lower activity on right-choice than left-choice trials, whereas neuron 2 has higher activity on right-choice than left-choice trials, leading to a choice-decoding direction $\vec{d}_{choice}$ that is orthogonal to $\vec{d}_{cue}$. Intuitively, if comparable proportions of sensory units are positively vs. negatively modulated by choice, the opposite signs of these modulations can cancel out when sensory unit activities are summed (projected onto $\vec{d}_{cue}$), leading to a readout of sensory information that is less confounded by internally generated choice signals. Our results are compatible with findings from areas MSTd

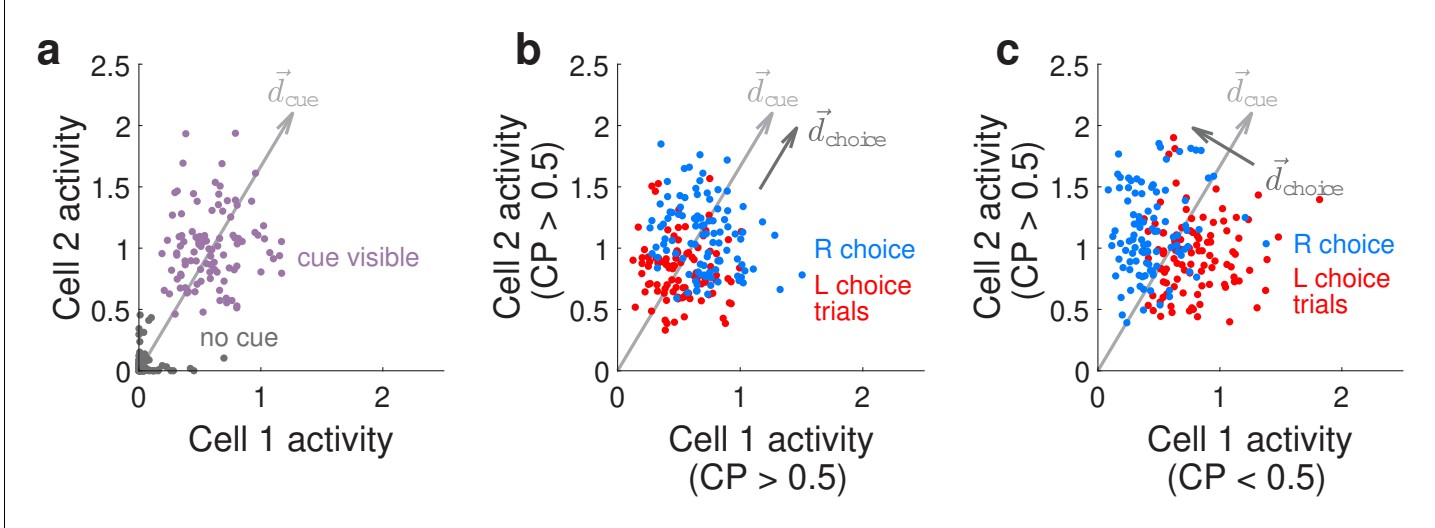

**Figure 5.** Conceptualization of how choice-related modulations can modify sensory representations at the neural-population level. (**a**) Illustrated distribution of the joint activity levels ('neural state') of two cue-locked cells, at time-points when there is no visual cue (dark gray), vs. time-points when a cue of the preferred laterality for these cells (purple) is present. Each time-point in this simulation corresponds to different samples of noise in the two neural responses, which results in variations in the neural state (multiple dots each corresponding to a different neural state). $\vec{d}_{cue}$ is a direction that best separates neural states for the 'no cue' vs. 'cue visible' conditions. (**b**) Illustrated distribution of neural states as in (**a**), but for time-points when a cue is present, colored differently depending on whether the mouse will eventually make a right-turn (blue) or left-turn choice. $\vec{d}_{choice}$ is a direction that best separates neural states for right- vs. left-choice conditions, which was chosen here to be parallel to $\vec{d}_{cue}$ (defined as in (**a**)). (**c**) Same as (**b**), but for a scenario where $\vec{d}_{choice}$ was chosen to be orthogonal to $\vec{d}_{cue}$.

and VIP of nonhuman primates that use alternative analyses (to CP) that more rigorously separates stimulus vs. choice effects on neural activity as they are behaviorally interrelated (**Zaidel et al., 2017**), as well as a recent re-analysis of area MT data in nonhuman primates performing an evidence-accumulation task (**Zhao et al., 2020**). Similar arguments have been made for how motor preparatory activity and feedback do not interfere with motor output (**Kaufman et al., 2014**; **Stavisky et al., 2017**), and how attentional-state signals can be distinguished from visual stimulus information (**Snyder et al., 2018**). The use of both positive and negative modulations for coding non-sensory information, such as choice here, may hint at a general coding principle that allows non-destructive multiplexing of information in the same neuronal population.

All in all, our neurophysiological observations in a mouse pulsatile evidence-accumulation task bears some similarity to but also notable differences with respect to an extensive body of related work on evidence accumulation tasks in nonhuman primates (NHP). We hypothesize that although a sensory evidence accumulation process may underlie the decision-making behaviors in all of these tasks, there are qualitative differences in both the nature of the tasks as well as our methods of investigation that may shed light on the differences in reported neurophysiological findings. From a methodological standpoint, the use of randomized pulsatile stimuli gives us the power to exploit the unpredictable (but known to the experimenter) timing of sensory pulses to separate stimulus responses from responses to other aspects of the behavior. One downside to this random design, together with the navigational nature of the task that the mouse controls, is that no two trials are literally identical. We thus trade off the richness of the behavior and the ability to directly identify sensory responses, with an inability to directly measure effects that require exactly repeated trials, such as noise correlations. The scope of our study should therefore be understood as being on signal responses across the posterior cortex, and we do not attempt here to report features such as noise correlations that are contingent on having a fully correct model of signal responses in order to interpret the residual as 'noise'.

Starting from our most basic neurophysiological observation, the small fractions of cue-locked neural activity in even the visual cortices is not unexpected, because the visual inputs of the task were not tuned to elicit maximal responses from the recorded neurons. In fact, the virtual spatial

environment that the mice experienced corresponds to a high rate of visual information beyond just the tower-like cues, all of which are highly salient visual inputs for performing the navigation aspect of the task and may therefore be expected to influence much of the activity in visual cortices. Our observation that choice-related variability in cue-locked cell responses were not lateralized according to brain hemisphere *Figure 4—figure supplement 2e*, is similar to the non-lateralized choice information in the activities of other (non-cue-locked) neurons that we report in an upcoming article (*Koay et al., 2019*). These findings of cells with intermixed choice preferences within the same brain hemisphere is compatible with several other rodent neurophysiological findings in evidence-accumulation tasks (*Erlich et al., 2011*; *Hanks et al., 2015*; *Scott et al., 2017*), but not, to the best of our knowledge, the NHP choice probability literature discussed above unless via alternative/extended analyses such as in *Zaidel et al., 2017*; *Zhao et al., 2020*. Other than analysis methodology and interspecies differences in brain architecture as a plausible cause for different neural representations of choice, we wonder if these differences could arise from choice and stimulus preferences being related in a more abstract way in our task (and other rodent behavioral paradigms) than in the NHP studies. In the Accumulating-Towers task, although the mouse should choose to turn to the side of the T-maze corresponding to the side with more cues, a navigational goal location is qualitatively different in modality from the retinotopic location of the tower-shaped cues. In contrast, in classic NHP evidence-accumulation tasks (*Gold and Shadlen, 2007*) the subject should saccade in the same direction as they perceive random dot motion stimulus to be along, that is perform a directly visual-direction-based action to indicate their choice. Our overall hypothesis is that if there is additional task-relevant information that have potentially abstract relationships to the visual cues to be accumulated, the brain may need to employ more complex neural representational schemes—including in as early as V1—in order to keep track of not only the momentary sensory information in visual cortices, but also various environmental and memory-based contexts in which they occur.

## Materials and methods

### Experiment subjects

All procedures were approved by the Institutional Animal Care and Use Committee at Princeton University (protocol 1910) and were performed in accordance with the Guide for the Care and Use of Laboratory Animals (*National Research Council, Division on Earth and Life Studies, Institute for Laboratory Animal Research, and Committee for the Update of the Guide for the Care and Use of Laboratory Animals, 2011*). We used 11 mice for the main experiments (+4 mice for control experiments), aged 2–16 months of both genders, and from three transgenic strains (see *Supplementary file 2*) that express the calcium-sensitive fluorescent indicator GCamp6f (*Chen et al., 2013*) in excitatory neurons of the neocortex:

- Six (+2 control) mice (6 male, 2 female): Thy1-GCaMP6f (*Dana et al., 2014*) [C57BL/6J-Tg (Thy1-GCaMP6f)GP5.3Dkim/J, Jackson Laboratories, stock # 028280]. Abbreviated as 'Thy1 GP5.3' mice.
- Five (+1 control) mice (three male, three female): Triple transgenic crosses expressing GCaMP6f under the CaMKIIα promoter, from the following two lines: Ai93-D; CaMKIIα-tTA [IgS5$^{tm93.1(tetO-GCaMP6f)Hze}$ Tg(Camk2atTA) 1Mmay/J (*Gorski et al., 2002*), Jackson Laboratories, stock #024108] (*Manita et al., 2015*); Emx1-IRES-Cre [B6.129S2-Emx1$^{tm1(cre)Krj}$/J, Jackson Laboratories, stock #005628]. Abbreviated as 'Ai93-Emx1' mice.
- One mouse (control experiments; female): quadruple transgenic crossexpressing GCaMP6f in the cytoplasm and the mCherry protein in the nucleus. both Cre-dependent, from the three lines: Ai93-D; CaMKIIα-tTA, Emx1-IRES-Cre, and Rosa26 LSL H2B mCherry [B6;129S-Gt(ROSA) 26Sor$^{tm1.1Ksvo}$/J, Jackson Laboratories, stock #023139].

Mice were randomly assigned such that there were about the same numbers of either gender and various transgenic lines in each group (main vs. control experiments). As the Ai93-Emx1 strain had higher expression levels of the fluorescent indicator, they produced significantly higher signal-to-noise (SNR) recordings than the Thy1 GP5.3 strain, and contributed more to the layer 5 datasets (see *Supplementary file 2*). Strain differences in the results were small and not of a qualitative nature (*Figure 2—figure supplement 1d–g*, *Figure 4—figure supplement 2h*).

## Surgery

Young adult mice (2–3 months of age) underwent aseptic stereotaxic surgery to implant an optical cranial window and a custom lightweight titanium headplate under isoflurane anesthesia (2.5% for induction, 1–1.5% for maintenance). Mice received one pre-operative dose of meloxicam subcutaneously for analgesia (1 mg/kg) and another one 24 hr later, as well as peri-operative intraperitoneal injection of sterile saline (0.5cc, body-temperature) and dexamethasone (2–5 mg/kg). Body temperature was maintained throughout the procedure using a homeothermic control system (Harvard Apparatus). After asepsis, the skull was exposed and the periosteum removed using sterile cotton swabs. A 5 mm diameter craniotomy approximately centered over the parietal bone was made using a pneumatic drill. The cranial window implant consisted of a 5 mm diameter round #1 thickness glass coverslip bonded to a steel ring (0.5 mm thickness, 5 mm diameter) using a UV-curing optical adhesive. The steel ring was glued to the skull with cyanoacrylate adhesive. Lastly, a titanium headplate was attached to the cranium using dental cement (Metabond, Parkell).

## Behavioral task

After at least three days of post-operative recovery, mice were started on water restriction and the Accumulating-Towers training protocol (*Pinto et al., 2018*), summarized here. Mice received 1–2 mL of water per day, or more in case of clinical signs of dehydration or body mass falling below 80% of the pre-operative value. Behavioral training started with mice being head-fixed on an 8-inch Styrofoam ball suspended by compressed air, and ball movements were measured with optical flow sensors. The VR environment was projected at 85 Hz onto a custom-built Styrofoam toroidal screen and the virtual environment was generated by a computer running the Matlab (Mathworks) based software ViRMEn (*Aronov and Tank, 2014*), plus custom code.

For historical reasons, 3 out of 11 mice were trained on mazes that were longer (30 cm pre-cue region + 250 cm cue region + 100–150 cm delay region) than the rest of the cohort (30 cm pre-cue region + 200 cm cue region + 100 cm delay region). In VR, as the mouse navigated down the stem of the maze, tall, high-contrast visual cues appeared along either wall of the cue region when the mouse arrived within 10 cm of a predetermined cue location; cues were then made to disappear after 200 ms (see following section for details on timing precision). Cue locations were drawn randomly per trial according to a spatial Poisson process with 12 cm refractory period between consecutive cues on the same wall side. The mean number of majority:minority cues was 8.5:2.5 for the 250 cm cue region maze and 7.7:2.3 for the 200 cm cue region maze. Mice were rewarded with $\geq 4\mu L$ of a sweet liquid reward (10% diluted condensed milk, or 15% sucrose) for turning down the arm on the side with the majority number of cues. Correct trials were followed by a 3s-long inter-trial-interval (ITI), whereas error trials were followed by a loud sound and an additional 9 s time-out period. To discourage a tendency of mice to systematically turn to one side, we used a de-biasing algorithm that adjusts the probabilities of sampling right- vs. left-rewarded trials (*Pinto et al., 2018*). Per session, we computed the percent of correct choices using a sliding window of 100 trials and included the dataset for analysis if the maximum performance was $\geq 65\%$.

## Functional identification of visual areas

We adapted methods (*Garrett et al., 2014*; *Kalatsky and Stryker, 2003*; *Zhuang et al., 2017*) to functionally delineate the primary and secondary visual areas using widefield imaging of calcium activity paired with presentation of retinotopic stimuli to awake and passively running mice. We used custom-built, tandem-lens widefield macroscopes consisting of a back-to-back objective system (*Ratzlaff and Grinvald, 1991*) connected through a filter box holding a dichroic mirror and emission filter. One-photon excitation was provided using a blue (470 nm) LED (Luxeon star) and the returning green fluorescence was bandpass-filtered at 525 nm (Semrock) before reaching a sCMOS camera (Qimaging, or Hamamatsu). The LED delivered about 2–2.5 mW/cm$^2$ of power at the focal plane, while the camera was configured for 20–30 Hz frame rate and about 5–10 μm spatial resolution. Visual stimuli were displayed on either a 32' AMVA LED monitor (BenQ BL3200PT), or the same custom Styrofoam toroidal screen as for the VR rigs. The screens were placed to span most of the visual hemifield on the side contralateral to the mouse's optical window implant. The space between the headplate and the objective was covered using a custom made cone of opaque material.

The software used to generate the retinotopic stimuli and coordinate the stimulus with the wide-field imaging acquisition was a customized version of the ISI package (*Juavinett et al., 2017*) and utilized the Psychophysics Toolbox (*Brainard, 1997*). Mice were presented with a 20° wide bar with a full-contrast checkerboard texture (25° squares) that inverted in polarity at 12 Hz, and drifted slowly (9°/s) across the extent of the screen in either of four cardinal directions (*Zhuang et al., 2017*). Each sweep direction was repeated 15 times, totaling four consecutive blocks with a pause in between. Retinotopic maps were computed similarly to previous work (*Kalatsky and Stryker, 2003*) with some customization that improved the robustness of the algorithms for preparations with low signal-to-noise ratios (SNR). Boundaries between the primary and secondary visual areas were detected using a gradient-inversion-based algorithm (*Garrett et al., 2014*), again with some changes to improve stability for a diverse range of SNR.

## Two-photon imaging during VR-based behavior

The virtual reality plus two-photon scanning microscopy rig used in these experiments follow a previous design (*Dombeck et al., 2010*). The microscope was designed to minimally obscure the ~270° horizontal and ~80° vertical span of the toroidal VR screen, and also to isolate the collection of fluorescence photons from the brain from the VR visual display. Two-photon illumination was provided by a Ti:Sapphire laser (Chameleon Vision II, Coherent) operating at 920 nm wavelength, and fluorescence signals were acquired using a 40 × 0.8 NA objective (Nikon) and GaAsP PMTs (Hamamatsu) after passing through a bandpass filter (542/50, Semrock). The amount of laser power at the objective used ranged from ~40–150 mW. The region between the base of the objective lens and the headplate was shielded from external sources of light using a black rubber tube. Horizontal scans of the laser were performed using a resonant galvanometer (Thorlabs), resulting in a frame acquisition rate of 30 Hz and configured for a field of view (FOV) of approximately $500 \times 500 \mu m$ in size. Microscope control and image acquisition were performed using the ScanImage software (*Pologruto et al., 2003*). Data related to the VR-based behavior were recorded using custom Matlab-based software embedded in the ViRMEn engine loop, and synchronized with the fluorescence imaging frames using the I2C digital serial bus communication capabilities of ScanImage. A single FOV at a fixed cortical depth and location relative to the functional visual area maps was continuously imaged throughout the 1–1.5 hr behavioral session. The vasculature pattern at the surface of the brain was used to locate a two-photon imaging FOV of interest.

## Identification of putative neurons

All imaging data were downsampled in time by a factor of 2 to facilitate analysis (i.e. 15 Hz effective frame rate), and first corrected for rigid brain motion by using the Open Source Computer Vision (OpenCV) software library function cv::matchTemplate. Fluorescence timecourses corresponding to individual neurons were then extracted using a deconvolution and demixing procedure that utilizes the Constrained Non-negative Matrix Factorization algorithm (CNMF [*Pnevmatikakis et al., 2016*]). A custom, Matlab Image Processing Toolbox (Mathworks) based algorithm was used to construct initial hypotheses for the neuron shapes in a data-driven way. In brief, the 3D fluorescence movie was binarized to mark significantly active pixels, then connected components of this binary movie were found. Each of these components arose from a hypothetical neuron, but a neuron could have contributed to multiple components. A shape-based matching procedure was used to remove duplicates before using these as input to CNMF. The 'finalized' components from CNMF were then selected post-hoc to identify those that resembled neural somata, using a multivariate classifier with a manual vetting step.

## General statistics

We summarize the distribution of a given quantity vs. areas and layers using quantile-based statistics, which are less sensitive to non-Gaussian tails. The standard deviation is computed as half the difference between the 84% and 16% quantiles of the data points. The standard error (S.E.M.) is computed as the standard deviation divided by $\sqrt{n}$ where $n$ is the number of data points. For uncertainties on fractions/proportions, we compute a binomial confidence interval using a formulation with the equal-tailed Jeffreys prior interval (*DasGupta et al., 2001*). The significance of differences

in means of distributions were assessed using a two-sided Wilcoxon rank sum test. The p-value threshold for evaluating significance is 0.05 for all tests, unless otherwise stated.

## Behavioral metrics

These analyses were described in a previous study (*Pinto et al., 2018*) and outlined here. The fraction of trials where a given mouse turned right was computed in 11 bins of evidence levels $\Delta \equiv \#R - \#L$ at the end of each trial, and fit to a 4-parameter sigmoid function $p_R(\Delta) = p_0 + B\left[1 + e^{-(\Delta - \Delta_0)/\lambda}\right]^{-1}$ to obtain psychometric curves. A logistic regression model was used to assess the dependence of the mice's choices on the spatial location of cues, that is, with factors being the evidence $\{\Delta_i | i = 1, 2, 3\}$ computed using cues in equally sized thirds of the cue region (indexed by $i$). Statistical uncertainties on the regression weights were determined by repeating this fit using 1000 bootstrapped pseudo-experiments.

## Precision of behavioral cue timings

The cue onset is defined as the instant at which a given cue is made visible in the virtual reality display, that is, when the mouse approaches 10 cm of the predetermined cue location in $y$, the coordinate down the stem of the maze. Given a typical mouse running speed of about 70 cm/s (*Figure 1—figure supplement 1a–b*) and the virtual reality display refresh rate of 85 Hz, there can be a lag of up to one frame (12 ms) or equivalently about 0.8 cm distance in the cue onset from the intended 10 cm approach definition. Regardless, the actual frame at which the cue appears was recorded in the behavioral logs and was used in all analyses.

The cues were made to vanish after 200 ms, but it is possible for a mouse to run so quickly that a given cue falls outside of the 270° virtual reality display range in less than 200 ms. Only 1/11 mice exhibited running speeds (~90 cm/s) that occasionally ran into this regime. *Figure 1—figure supplement 1c–d* shows that for 10/11 mice the actual duration of visibility of cues was essentially 200 ms (standard deviation <10 ms), while for the one fast mouse the cue duration was ~190 ms (standard deviation <30 ms).

We do not expect cue-responsive neurons in the visual cortices to continue responding strongly to a given cue for the entire 200 ms for which it is visible, because of the expected retinotopy of visual cortical responses and previous reports of 15°−20° receptive field radii. For a neuron with a 20° receptive field that has one edge at the cue onset location (10 cm ahead and 4 cm lateral of the mouse), the cue would fall outside of a 40° diameter receptive field within 130 ms (110 ms) if the mouse ran straight past it at 60 cm/s (70 cm/s). In sum, we expect the variability in how long a cue remains in neural receptive fields to be on the order of 10 s of milliseconds (or less for neurons with more lateralized receptive fields).

## Impulse response model for cue-locked cells

This analysis excluded some rare trials where the mouse backtracks through the T-maze, by using only trials where the $y$ displacement between two consecutive behavioral iterations was $> -0.2$ cm (including all time-points up to the entry to the T-maze arm), and if the duration of the trial up to and not including the ITI was no more than 50% different from the median trial duration in that session.

We modeled the activity of each cell as a time series of non-negative amplitudes $A_i$ in response to the $i^{th}$ cue, convolved with a parametric impulse response function $g(t)$:

$$\frac{\Delta F}{F}(t) = \sum_{i=1}^{m} A_i \, g\left(t - t_i - \tau_{\text{lag}} - \delta\tau_i; \sigma_\uparrow, \sigma_\downarrow\right) + \text{i.i.d.noise}$$

$$g\left(t; \sigma_\uparrow, \sigma_\downarrow\right) = \left[\frac{\sqrt{2/\pi}}{\sigma_\uparrow + \sigma_\downarrow} \begin{cases} e^{-t^2/2\sigma_\uparrow^2}, & t < 0 \\ e^{-t^2/2\sigma_\downarrow^2}, & t \geq 0 \end{cases}\right] * h_{\text{Ca}^{2+}}(t) \qquad (1)$$

where $\{t_i | i = 1, \ldots, m\}$ are the appearance times of cues throughout the behavioral session. The free parameters of this model are the lag ($\tau_{lag}$), rise ($\sigma_\uparrow$) and fall ($\sigma_\downarrow$) times of the impulse response function, the amplitudes $A_i$, and small (L2-regularized) time jitters $\delta\tau_i$ that decorrelates variability in response timings from amplitude changes. $h_{Ca^{2+}}(t)$ is a calcium indicator response function using

parameters from literature (*Chen et al., 2013*), which deconvolves calcium and indicator dynamics from our reports of timescales. This function is parameterized as a difference of exponentials, $h_{Ca^{2+}}(t) = \left(1 - e^{-t/\tau_{\uparrow}^{Ca}}\right) e^{-t/\tau_{\downarrow}^{Ca}}/h_0$, where $\tau_{\uparrow}^{Ca} \equiv 35ms$, $\tau_{\downarrow}^{Ca} \equiv 300ms$, and $h_0$ is a normalization constant such that the peak of this function is 1. The per-cue time jitter parameters $\delta\tau_i$ additionally allows this model to flexibly account for some experimental uncertainty in the assumed cue onset times (see previous section). The distribution of jitter parameters that we obtained from fitting this model (as explained below) had a standard deviation of about 50 ms across neurons, which is within the expected range of behavioral timing variations. We also note that neural response timescales cannot be resolved to better than the Nyquist rate of the imaging data, $(1/15Hz)/2 \approx 33ms$.

We maximized the model likelihood to obtain point estimates of all the parameters, using a custom coordinate-descent-like algorithm (*Wright, 2015*). The significance of a given cell's time-locking to cues was defined as the number of standard deviations that the impulse response model $AIC_C$ score (bias-corrected Aikaike Information Criterion [*Hurvich and Tsai, 1989*]) lies above the median $AIC_C$ of null hypothesis models where the timings $\{t_i\}$ of cues were randomly shuffled within the cue region. Given the $\Delta F/F$ time-series $F(t)$ for a given cell and the predicted activity time-trace $m(t)$ which we treat as vectors $\vec{F}$ and $\vec{m}$ respectively, the $AIC_C$ score is:

$$\mathrm{AICC}\left(\vec{F}, \vec{m}\right) := 2n_{\mathrm{par}} + n_F \ln\left(\frac{\left\|\vec{F} - \vec{m}\right\|^2}{n_F}\right) + 2n_{\mathrm{par}}\frac{(n_{\mathrm{par}}+1)}{(n_F - n_{\mathrm{par}} - 1)}$$

where $n_F$ is the number of time-points that comprise the data and $n_{par}$ is the number of free parameters in the model. Lastly, a small fraction of cells responded to both left- and right-side cues. We parsimoniously allowed for different impulse responses to these by first selecting a primary response (preferred-side cues) as that which yields the best single-side model $AIC_C$, then adding a secondary response if and only if it would improve the model likelihood. This criterion is $\exp\left(\left[\mathrm{AICC}\left(\vec{F}, \vec{m}_{(1)}\right) - \mathrm{AICC}\left(\vec{F}, \vec{m}_{(1)} + \vec{m}_{(2)}\right)\right]/2\right) \geq 0.05$, where $\vec{m}_{(1)}$ is the model prediction with only primary responses and $\vec{m}_{(1)} + \vec{m}_{(2)}$ is the model prediction with both primary and secondary responses. We defined cells to be cue-locked if the primary response significance exceeded three standard deviations of the abovementioned null hypotheses. Other than a factor of about two reduction in the number of identified cue-locked neurons, we found no qualitative difference in our conclusions for a much stricter significance threshold of 5 standard deviations.

## Decoding from cue-locked amplitudes

The decoding models were fit separately using responses to cues in three equally sized spatial bins of the cue region. We defined the neural state response as the vector of contralateral-cue-locked cell response amplitudes to a given cue, and used a Support Vector Machine classifier (SVM) to predict a task variable of interest from this neural state (using data across trials but restricted to responses to cues in a given third of the cue region, as mentioned). To assess the performance of these classifiers using threefold cross-validation, we trained the SVM using 2/3rds of the data and computed Pearson's correlation coefficient between the predicted and actual task variable values in the held-out 1/3rd of the data. Significance was assessed by constructing 100 null hypothesis pseudo-experiments where the neural state for a given epoch bin was permuted across trials, that is preserving inter-neuron correlations but breaking any potential relationship between neural activity and behavior.

To correct for multiple comparisons when determining whether the decoding p-value for a particular dataset was significant, we used the Benjamini-Hochberg procedure (*Benjamini and Hochberg, 1995*) as follows. For a given type of decoder, we sorted the *p*-values of all data points (spatial bins and imaging sessions) in ascending order, $[p_1, p_2, ..., p_n]$, and found the first rank $i_\alpha$ such that $p_{i_\alpha} \leq i_\alpha \times 0.05/n$. The decoding performance was then considered to be significantly above chance for all $p \leq p_{i_\alpha}$.

## Uncorrelated modes of task variables

We wished to define a set of uncorrelated behavioral modes such that the original set of six task variables are each a linear combination of these modes, with the additional requirement that each mode should be as similar as possible to one of the task variables. In matrix notation, this means that we want to solve:

$$\underset{\mathbf{Y}}{\mathrm{argmin}} \, \|\mathbf{Y} - \mathbf{X}\|_F \quad \text{s.t.} \quad \mathbf{Y}^\top \mathbf{Y} = \mathbf{I}$$

where each column of $\mathbf{X}$ corresponds to values of a given task variable across trials, each column of $\mathbf{Y}$ are the uncorrelated behavioral modes, and $\|\mathbf{A}\|_F \equiv \left( \sum_{ij} A_{ij}^2 \right)^{1/2}$ is the Frobenius norm of a matrix $\mathbf{A}$. This can be computed using polar decomposition (*Higham, 1988*): $\mathbf{X} = \mathbf{Y}\mathbf{H}$, where $\mathbf{Y}$ is an orthogonal matrix and $\mathbf{H}$ a symmetric matrix. To obtain the polar decomposition, we used an algorithm based on the singular value decomposition $\mathbf{X} = \mathbf{U}\mathbf{\Sigma}\mathbf{V}^\top$, which gives the solution $\mathbf{Y} = \mathbf{U}\mathbf{V}^\top$.

## Amplitude modulation models

These models used as input the following behavioral data: $t_i$ is the onset time of the $i^{th}$ cue, which is located at distance $y_i$ along the cue region, and appears at a visual angle $\phi_{cue}(t_i)$ relative to the mouse (*Figure 1c*). $\Delta(t_i)$ is the cumulative cue counts (explained further below) up to and including cue $i$, and $C(t_i)$ is the upcoming choice of the mouse in that trial. $v_i \equiv v(t_i)$ is the running speed of the mouse in the virtual world at the time that the $i^{th}$ cue appeared, and for simple linear speed dependencies explained below, the standardized version $\tilde{v}(t) \equiv \left[ v(t) - Q_{50\%}^v \right] / \left[ Q_{90\%}^v - Q_{10\%}^v \right]$ is used, where $Q_p^v$ is the $p$ probability content quantile of the speed distribution.

To account for the stochastic and nonnegative nature of pulsatile responses, the cue-locked cell response amplitudes $A_i$ were modeled as random samples from a Gamma distribution, $A_i \mid \mu_A(t_i), k \sim \Gamma[k, \mu_A(t_i)/k]$. The shape parameter $k$ for the Gamma distribution is a free parameter, and furthermore indexed by choice for the choice model. The four models discussed in the text are defined by having different behavior-dependent mean functions $\mu_A(t_i)$ that have the following forms (detailed below):

$$\mu_A(t_i) = \rho[\phi_{cue}(t_i)] \times \begin{cases} s_v[v(t_i)] & \text{null hypothesis} \\ [1 + \varphi \tilde{v}(t_i)] h[\mu_A(t_{i-1}), t_i - t_{i-1}] & \text{SSA} \\ [1 + \varphi_c \tilde{v}(t_i)] s_y^c(y_i) & \text{choice} \\ [1 + \varphi \tilde{v}(t_i)] s_\Delta[\Delta(t_i)] & \text{cue-counts} \end{cases} \quad (2)$$

In all of the models, $\rho(\phi_{cue})$ is an angular receptive field function that has either a skew-Gaussian (*Priebe et al., 2006*) or sigmoidal dependence on $\phi_{cue}$:

$$\rho(\phi) = \begin{cases} \exp\left( -\frac{1}{2} \frac{(\phi - \phi_0)^2}{[\sigma + \zeta(\phi - \phi_0)]^2} \right) - \exp\left( -\frac{1}{\zeta^2} \right) & \text{skew} - \text{Gaussian} \\ \dfrac{1 - \rho_0}{2} + \dfrac{\rho_0}{[1 + exp(-[\phi - \phi_0]/\zeta)]^\nu} & \text{sigmoid} \end{cases}$$

$\phi_0, \sigma, \zeta, \rho_0$ and $\nu$ are all free parameters, and either the skew-Gaussian or sigmoidal hypotheses are selected depending on which produces a better fit for the cell (using the $AIC_C$ score as explained below).

All the models also have a speed dependence that multiplies the angular receptive field function. For the null hypothesis, we allowed this to be highly flexible so as to potentially match the explanatory power of the other models (which have other behavioral dependencies). Specifically, the function $s_v(v)$ is defined to be a cubic spline (piecewise 3rd-order polynomial [*Gan, 2004*]) with control points at five equally-spaced quantiles of the running speed distribution, that is, at $v = \{Q_0^v, Q_{25\%}^v, Q_{50\%}^v, Q_{75\%}^v, Q_{100\%}^v\}$. A cubic spline model has as many free parameters as the number of control points. For the other models, we used a simple linear parameterization for speed dependence, $1 + \psi \tilde{v}$ where $\psi$ is a free parameter (for the choice model, there are two free parameters $\psi_c$ where $C$ indexes the choice).

The SSA, choice, and cue-counts models are further distinguished by how they depend on the $h$, $s_y^c$, and $s_\Delta$ functions respectively. For the SSA model:

$$h[\mu_A(t_{i-1}), t_i - t_{i-1}] = 1 + [\xi\, \mu_A(t_{i-1}) - 1] \exp[-(t_i - t_{i-1})/\lambda]$$

The response to the first cue in the session is defined to be $\mu_A(t_1) = 1$. The $h$ function can be understood as follows. Right after the cue at $t_{i-1}$, the response is scaled by the free parameter $\xi$, that is, the new response level is $\xi\, \mu_A(t_{i-1})$ where $\xi > 1$ corresponds to facilitation and corresponds to depression. This facilitation/depression effect decays exponentially with time toward 1, that is, the amount by which the response $\mu_A(t_i)$. deviates from one is equal to the deviation (from 1) of the facilitated/depressed response $\xi\, \mu_A(t_{i-1}) - 1$, multiplied by the time-recovery factor $\exp[-(t_i - t_{i-1})/\lambda]$. Here $\lambda$ is another free parameter that specifies the timescale of recovery.

The choice model has smooth dependencies on $y$ location on the cue region parameterized by choice. This is given by two functions $s_y^c(y)$ where $C$ indexes either the right or left choice, and each of these functions is a cubic spline with control points at $y = \{0, 0.5 L_{\text{cue}}, L_{\text{cue}}\}$ (recall that $L_{\text{cue}}$ is the total length of the cue region).

Lastly, the cue-counts model also has smooth dependencies on cue counts $\Delta$, that is, the function $s_\Delta(\Delta)$ is a cubic spline. As the responses of cells can depend on counts on either the right, left, or both sides (*Scott et al., 2017*), we allowed $\Delta$ to be either the cumulative right or cumulative left cue counts (control points are at $\Delta = \{0, 3, 8\}$), or the cumulative difference $\#R - \#L$ in cue counts (control points are at $\Delta = \{-4, 0, 4\}$). The best definition of $\Delta$ was selected per cell according to which produced the best $\text{AIC}_C$ score.

Because neural activity can be very different in the rare cases where the mouse halts in the middle of the cue region, only data where the speed $v$ is within 25% of its median value were included in the analysis of this model. Point estimates for the model parameters were obtained by minimizing the Gamma-distribution negative log-likelihood:

$$-\ln L = \sum_i (1-k) \ln A_i + \frac{A_i}{\mu_A(t_i)} + k \ln \mu_A(t_i) + \ln \Gamma(k)$$

Because the Gamma distribution is defined only in the positive domain, we had to make an assumption about how to treat data points where $A_i = 0$. We reasoned that we could substitute these with a noise-like distribution of amplitudes, which were obtained by fitting the impulse response model (*Equation 1*) using the same cue timings but simulated noise-only data, which comprised of a $\Delta F/F$ time-series drawn i.i.d. from a Gaussian distribution with zero mean and standard deviation being $\sigma^F$, the estimated fluorescence noise level for that cell. The relative $\text{AIC}_C$-based likelihood used for model selection as described in the text, is $\exp([\text{AICC}(\text{model 1}) - \text{AICC}(\text{model 2})]/2)$.

## Choice modulation strength

The location-dependent choice modulation strength for cue-locked amplitudes is defined as $\delta A^{\text{choice}}(y) = \left[ A_{\text{contra}}^{\text{choice}}(y) - A_{\text{ipsi}}^{\text{choice}}(y) \right] \langle A \rangle$, where $A_{\text{contra}}^{\text{choice}}(y) \equiv \mu_A(y; \ C = \text{contralateral choice})$ as in *Equation 2*, and analogously for ipsilateral choices. This is computed by evaluating the amplitude model prediction vs. location in the cue region, but at fixed $\phi_{\text{cue}}$ corresponding to zero view angle ($+22°$ for right-side cues and $-22°$ for left-side cues) and $\Delta = 0$. The normalization constant is:

$$\langle A \rangle = \left( \frac{1}{2 L_{\text{cue}}} \sum_{C \in \{R,L\}} \int_0^{L_{\text{cue}}} \frac{1}{max\left[ A_C^{choice}(y), \sigma^F \right]}\, dy \right)^{-1}$$

## Acknowledgements

We thank BB Scott for brainstorming and feedback on the concept of this paper, as well as L Pinto, CM Constantinople, AG Bondy, M Aoi, and B Deverett for useful and interesting discussions. B Engelhard and L Pinto built rigs for the high-throughput training of mice, and S Stein helped in the training of mice in this study. B Engelhard and L Pinto contributed behavioral data from the mouse evidence accumulation task. We additionally thank all members of the BRAIN COGS team, Tank and

Brody labs. This work was supported by the NIH grants 5U01NS090541 and 1U19NS104648, and the Simons Collaboration on the Global Brain (SCGB).

## Additional information

### Funding

| Funder | Grant reference number | Author |
|---|---|---|
| National Institutes of Health | 5U01NS090541 | Sue Ann Koay<br>Stephan Thiberge<br>Carlos D Brody<br>David W Tank |
| National Institutes of Health | 1U19NS104648 | Sue Ann Koay<br>Stephan Thiberge<br>Carlos D Brody<br>David W Tank |
| Simons Foundation | Simons Collaboration on the Global Brain-328057 | Carlos D Brody<br>David W Tank |

The funders had no role in study design, data collection and interpretation, or the decision to submit the work for publication.

### Author contributions

Sue Ann Koay, Conceptualization, Data curation, Software, Formal analysis, Investigation, Methodology, Writing - original draft; Stephan Thiberge, Resources, Supervision; Carlos D Brody, Conceptualization, Supervision, Writing - review and editing; David W Tank, Conceptualization, Resources, Supervision, Writing - review and editing

### Author ORCIDs

Sue Ann Koay (ID) https://orcid.org/0000-0002-9648-2475
Stephan Thiberge (ID) https://orcid.org/0000-0002-6583-6613
Carlos D Brody (ID) https://orcid.org/0000-0002-4201-561X
David W Tank (ID) https://orcid.org/0000-0002-9423-4267

### Ethics

Animal experimentation: All procedures were approved by the Institutional Animal Care and Use Committee at Princeton University (Protocol 1910) and were performed in accordance with the Guide for the Care and Use of Laboratory Animals (National Research Council et al. 2011). All surgeries were performed under isoflurane anesthesia, every effort was made to minimize suffering, and all experimental animals were group housed in enriched environments.

### Decision letter and Author response

Decision letter https://doi.org/10.7554/eLife.60628.sa1
Author response https://doi.org/10.7554/eLife.60628.sa2

## Additional files

### Supplementary files

• Supplementary file 1. Number of imaging sessions and mice for various areas and layers, for the main experiment.

• Supplementary file 2. Overall performance and number of imaging sessions for the main experiment, per mouse (rows), in various areas and layers (columns). Mice of the Thy1 GP5.3 strain have names starting with 'gp', and those from the Ai93-Emx1 strain have names starting with 'ai' (see Materials and methods).

• Transparent reporting form

## Data availability

A condensed set of imaging and behavioral data as well as secondary results from analyses and modeling have been deposited in Dryad with the DOI: https://doi.org/10.5061/dryad.tb2rbnzxv. This dataset contains all of the information required to reproduce the figures in the manuscript. As the full, raw data generated in this study is extremely large, access to these raw data can be arranged upon reasonable request to the authors.

The following dataset was generated:

| Author(s) | Year | Dataset title | Dataset URL | Database and Identifier |
|---|---|---|---|---|
| Koay SA, Thiberge SY, Brody CD, Tank DW | 2020 | Amplitude modulations of cortical sensory responses in pulsatile evidence accumulation | http://dx.doi.org/10.5061/dryad.tb2rbnzxv | Dryad Digital Repository, 10.5061/dryad.tb2rbnzxv |

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
