## [Decision Letter]

**Acceptance summary:**

This study investigates how sensory representations in visual cortex are modulated by ongoing task requirements as rats navigate a virtual environment and make a choice based on the total numbers of discrete stimuli, or 'pulses,' seen along the path. The main finding is that only a small fraction of active neurons had sensory-like responses time-locked to each pulse, and furthermore, for those that did, the amplitude of the response changed systematically as the impending choice advanced to completion. This shows that, even at a very basic level, the representation of sensory stimuli is strongly modulated and shaped by cognitive factors and behavioral relevance, and that a lot of the variability associated with sensory activity is not just random noise, as it often appears.

**Decision letter after peer review:**

Thank you for submitting your article "Amplitude modulations of sensory responses, and deviations from Weber's Law in pulsatile evidence accumulation" for consideration by *eLife*. Your article has been reviewed by three peer reviewers, one of whom is a member of our Board of Reviewing Editors, and the evaluation has been overseen by Michael Frank as the Senior Editor. The reviewers have opted to remain anonymous.

The reviewers have discussed the reviews with one another and the Reviewing Editor has drafted this decision to help you prepare a revised submission.

We would like to draw your attention to changes in our revision policy that we have made in response to COVID-19 (https://elifesciences.org/articles/57162). Specifically, when editors judge that a submitted work as a whole belongs in *eLife* but that some conclusions require additional new data or analyses, as they do with your paper, we are asking that the manuscript be revised to either limit claims to those supported by data in hand, or to explicitly state that the relevant conclusions require additional supporting data.

Our expectation is that the authors will eventually carry out the additional work and report on how it affects the relevant conclusions either in a preprint on bioRxiv or medRxiv, or if appropriate, as a Research Advance in *eLife*, either of which would be linked to the original paper.

Summary:

This manuscript carefully studies the properties of sensory responses in several visual areas during performance of a task in which head-fixed mice run along a virtual corridor and must turn toward the side that has more visual cues (small towers) along the wall. The results provide insight into the mechanisms whereby sensory evidence is accumulated and weighted to generate a choice, and on the sources of variability that limit the observed behavioral performance. All reviewers thought the work was generally interesting, carefully done, and novel.

However, the reviewers' impression was that the manuscript as it stands is very dense. In fact, it is largely two studies with different methods and approaches rolled into one. The first one (physiology) is still dense but less speculative and with interesting, solid results, and the revisions suggested by the reviewers should be relatively straightforward to address. In contrast, the modeling effort is no doubt connected to the physiology, but it really addresses a separate issue. The general feeling was that this material is probably better suited for a separate, subsequent article, for two reasons. First, because it will require substantial further work (see details below), and second, because it adds a fairly complex chapter to an already intricate analysis of the neurophysiological data.

So, going forward, we suggest that the authors revise the neurophys analyses along the lines suggested below (largely addressing clarity and completeness), leaving out the modeling study for a later report. If, however, the authors wish to maintain the current structure, they should address all the comments, and understand that we would reconsider the manuscript's suitability for publication after full re-review.

1) More should be done to highlight how very different the sensory representation is in this study compared with the great majority of earlier related work in the primate. This merits at least some discussion, and optimally, additional analyses of correlations in the data. See the comments from reviewer 3 for details.

2) Figure 4E was confusing. What is the point of showing the shades (which extend very far)? If the idea is to contrast the SSA and feedback models, then it would be better to plot their corresponding effects directly, on the same graph, or to show predictions versus actual data in each case, in two graphs. In any case, the data need to be shown in a different way, or the point made differently.

Similarly for Figure 3F. Could the authors explain how each point is calculated? I was specially confused about the meaning of the points *for each area* in the x-axis.

3) The prediction about the Fano Factor (FF) is problematic in a couple of ways. First, it seems to come out of the blue because Figure 5 is described before any discussion of the variability in the model is presented (except for the dice in the model schematic).

And second, the FF prediction itself is verified for a very small fraction of neurons even when an unusual pValue of 0.1 is used. Furthermore, the mathematical derivation relies on an Taylor series around *N_R_* ~ 0? (In most of the paper, CLT is invoked on the assumption *N_R_* is "large"). Due to the lack of transparency of this prediction and the mild support, the authors could consider dropping it, at least from the main manuscript.

4). The 1st prediction in Figure 5 seems very unspecific. In particular, it would seem like any "open loop" modulation of the cue-locked response which depended on time, or on location along the track, would induce a trend like the one assessed in Figure 5C-E. It is not clear this is a prediction specific to the multiplicative-feedback model the authors are advocating for.

5) The number of cells showing responses consistent with the model (Figure 5E) seems very small (~15% of 5-10% of cells with cue-locked responses). Could they really underlie the behavioral effects? The authors could perhaps comment on this.

6) It wasn't completely clear how the time of a particular cue onset was defined. In a real environment the cues would appear small (from afar) and get progressively bigger as the animal advances (at least if they are 3D objects, as depicted in Figure 1). What would be the cue onset in that case, and does the virtual environment work in the same way? This is probably not a serious issue, but it comes across as a bit at odds with the supposed "pulsatile" nature of the sensory stream, and would seem somewhat different from the auditory case with clicks.

A related question concerns multiple references to cue timing made in the Introduction, as if such timing were very precise. This seems strange given that all time points depend on the running speed of the mice, which is surely variable. So, how exactly is cue position converted to cue time, and why is there an assumption of very low variability? Some of this detail may be in previous reports, but it would be important to make at least a brief, explicit clarification early on.

Revisions expected in follow-up work:

For details, see comments 1-3 from reviewer 2 and comment 1 from reviewer 1, below.

Reviewer #1:

This study investigates the responses of neurons in the parietal cortex of mice (recorded via two-photon Ca imaging) performing a virtual navigation task, and then relates their activity to the animal's psychophysical performance. It is essentially two studies rolled into one. The analysis of neurophysiological activity in the first part shows that visually driven responses in the recorded "cue cells" are strongly modulated by the eventual choice and/or by the integrated quantity that defines that choice (the difference in left vs. right stimulus counts), as well as by other task variables, such as running speed. The model comparison study of the second part shows that, in the context of a sensory-motor circuit for performing the task, this type of feedback may account for subtle but robust psychophysical effects observed in the mice from this study and in rats from previous studies from the lab. Notably, the feedback explains intriguing deviations in choice accuracy from the Weber-Fechner law.

Both parts are interesting and carefully executed, although both are pretty dense; there are a ton of important technical details at each step. I wonder if this isn't too much for a single study. Had I not been reading it as a reviewer, I probably would have stopped after Figure 4 or just skimmed the rest. After that, the motivation, methods, and analyses shift markedly. I'm not pushing hard on this issue, but I think the authors should ponder it.

Other comments:

1) Figure 6 and the accompanying section of the manuscript investigate a variety of models with different architectures (feedback vs. purely feedforward) and noise sources. Here, if I understood correctly, the actual cue-driven responses are substituted with variables that are affected by different types of noise. It is this part that I found a bit disconnected from the rest, and somewhat confusing.

Here, there's a jump from the actual cells to model responses. I think this needs an earlier and more explicit introduction. It is clear what the objective of the modeling effort is; what's unclear are the elements that initially go into it. This is partly because the section jumps off with a discussion about accumulator noise, but the modeling involves many more assumptions (i.e., simplifications about the inputs to the accumulators).

What I wondered here was, what happened to all the variance that was carefully peeled away from the cue driven responses in the earlier part of the manuscript? Were the dependencies on running speed, viewing angle, contra versus ipsi sensitivity, etc still in play, or were the modeled cue-driven responses considering just the sensory noise from the impulse responses? I apologize if I missed this. I guess the broader question is how exactly the noise sources in the model relate to all the dependencies of the cue cells exposed in the earlier analyses.

Overall, my general impression is that this section requires more unpacking; perhaps it should become an independent report.

Reviewer #2:

In this manuscript, the authors present an in-depth analysis of the properties of sensory responses in several visual areas during performance of an evidence-accumulation task for head-fixed running mice (developed and studied by the authors previously), and of how these properties can illuminate aspects of the performance of mice and rats during pulsatile evidence accumulation, with a focus on the effect of "overall stimulus strength" on discriminability (Weber-Fechner scaling).

The manuscript is very dense and presents many findings, but the most salient ones are a description of how the variability in the large Ca++ transients evoked by the behaviourally-relevant visual stimuli (towers) are related to several low-level behavioural variables (speed, view) and also variables relevant for the task (future choice, running count of accumulated evidence), and a framework based on multiplicative-top down feedback that seeks to explain some aspects of this variability and ultimately the psychophysical performance in the accumulating-towers task. The first topic is framed in the context of the literature on choice-probability, and the second in the context of "Weber-Fechner" scaling, which in the current task would imply constant performance for given ratios of Left/Right counts as their total number is varied.

Overall, the demonstration of how trial to trial variability is informative about various relevant variables is important and convincing, and the model with multiplicative feedback is elegant, novel, naturally motivated by the neural data, and an interesting addition to a topic with a long-history.

1) Non-integrable variability. In addition to 'sensory noise' (independent variability in the magnitude of each pulse), it is critical in the model to include a source of variability whose impact does not decay through temporal averaging (to recover Weber-Fechner asymptotically for large N). This is achieved in the model by positing trial-to-trial variability (but not within-trial) in the dot product of the feedforward (w) and feedback (u) directions. But the way this is done seems to me problematic:

The authors model variability in w*u as LogNormal (subsection “Sources of noise in various accumulator architectures”). First, the justification for this choice is incorrect as far as I can tell. The authors write: "We model m^R with a lognormal distribution, which is the limiting case of a product of many positive random variables". But neither is the dot product of w and u a product (it's a sum of many products), nor are the elements of this sum positive variables (the vector u has near zero mean and both positive and *negative* elements allowing different neurons to have opposite preferences on choice – see e.g., in the subsection “Cue-locked amplitude modulations motivate a multiplicative feedback-loop circuit model” where it is stated that *u_i_*<0 for some cells), nor would it have a LogNormal distribution even if the elements of the sum were indeed positive. Without further assumptions, the dot product w*u will have a normal distribution with mean and variance dependent on the (chosen) statistics of u and w.Two conditions seem to be necessary for u*w: it should have a mean positive but close to zero (if it's too large a(t) will explode), and it should have enough variability to make non-integrable noise have an impact in practice. For a normal distribution, this would imply that for approximately half of the trials, w*u would need to be negative, meaning a decaying accumulator and effectively no feedback. This does not seem like a sensible strategy that the brain would use.

The authors should clarify how this LogNormality is justified and whether it is a critical modelling choice (as an aside, although LogNormality in u*w allows non-negativity, low mean and large variability, the fact that it has very long tails sometimes leads to instability in the values of a(t)).

2) Related to this point, it would be helpful to have more clarity on exactly what is being assumed about the feedback vector u. The neural data suggests u has close to zero mean (across neurons). At the same time, it is posited that u varies across trials ("accumulator feedback is noisy") is and that this variability is significant and important (previous comment). However, it would seem like neurons keep their choice preference across trials, meaning the trial to trial variability in each element of u has to be smaller than the mean. The authors only describe variability in u*w (LogNormal), but, in addition to the issues just mentioned about this choice, what implications does this have for the variability in u? The logic of the approach would greatly increase if the authors made assumptions about the statistics of u consistent with the neural data, and then derived the statistics of u*w.

3) Overall, it seems like there is an intrinsically hard problem to be solved here, which is not acknowledged: how to obtain large variability in the effective gain of a feedback loop while at the same time keeping the gain "sufficiently restricted", i.e., neither too large and positive (runaway excitation) nor negative (counts are forgotten). While the authors avoid worrying about model parameters by fitting their values from data (with the caveats discussed above), their case would become much stronger if they studied the phenomenology of the model itself, exposing clearly the computational challenges faced and whether robust solutions to these problems exist.

Reviewer #3:

This manuscript describes measurements of neuronal activity in mice performing a discrimination task, and a new model that links these data to psychophysical performance. The key element of the new model is that sensory neurons are subject to gain modulations that evolve during each trial. They show that the model can produce pure sensory integration, Weber-Fechner performance, or intermediate states that nicely replicate the behavioral observations. This is an interesting and valuable contribution.

My only significant comment relates to the Discussion, which should do more to make sure the reader understands how very different the sensory representation is in this study compared with the great majority of earlier related work in the primate:

First, choice related signals are not systematically related to stimulus preferences (no Choice Probability). This is mentioned, but only very briefly.

Second, there appears to be no relationship between stimulus preference (visual field in this case) and noise correlation. Unfortunately, this emerges from the model fits, not an analysis of data. But is an important difference with profound implications for how the coding of information is organized. It really needs a discussion. It should also be supported by an analysis of correlations in the data. I know some people argue that 2 photon measures make this difficult, but if that's true then surely they can’t be used to support a model in which correlations are a key component.

---

## [Author Response]

To our understanding, items 3-5 listed in this section of the decision letter are only relevant for the accumulator modeling work, and we have therefore moved them to the next section.

1) More should be done to highlight how very different the sensory representation is in this study compared with the great majority of earlier related work in the primate. This merits at least some discussion, and optimally, additional analyses of correlations in the data. See the comments from reviewer 3 for details.

We have replaced the last half of the Discussion, which used to be about the accumulator circuit models, with an extended discussion of how the neural responses to visual cues in our task relate to and differ from previous work on nonhuman primates (NHP) and rodents. Most of this discussion concerns points brought up by reviewer 3 (please see the specific reply to reviewer 3 below for details). In particular, we discuss how the subject’s eventual choice modifies sensory representations, which we illustrate in an added conceptual Figure 5. The last Discussion paragraph provides a more general comparison between our work vs. other rodent work and the NHP literature. With respect to reviewer 3’s request for analyses of correlations in the data, we have refrained from doing so because we do not think that we can make correct claims about noise correlations in our data. The reason is the nature of the behavior which could not have truly repeated trials. Please see the reply to reviewer 3 for details.

2) Figure 4E was confusing. What is the point of showing the shades (which extend very far)? If the idea is to contrast the SSA and feedback models, then it would be better to plot their corresponding effects directly, on the same graph, or to show predictions versus actual data in each case, in two graphs. In any case, the data need to be shown in a different way, or the point made differently.

We have replaced Figure 4E in the revised manuscript to provide a direct comparison of the neural vs. behavioral timecourses. We wanted the timecourse of neural choice modulations in Figure 4E to be compared to the timecourse of how cues influenced the behavioral performance data in Figure 1E. Perhaps because these panels were separated by many figures, it is not obvious what the reader should take away by the time they come to Figure 4E.

Similarly for Figure 3F. Could the authors explain how each point is calculated? I was specially confused about the meaning of the points for each area in the x-axis.

We have added a more detailed explanation of the computation of Figure 3F to both the text and the caption, as follows. The goal of Figure 3F is to address the visible differences across brain areas, in Figure 3B, D, in how well various task variables could be decoded from the amplitudes of a population of cue-locked cells. We wanted to know if these differences were indeed region-specific differences or whether they could be explained by differences in the number of recorded neurons (which differed systematically across cortical areas/layers). To do this we constructed a linear regression model to predict the decoding performance (evaluated in the middle of the cue region for each dataset) as a weighted sum of a set of factors being the x-axis coordinates in Figure 3F. The cortical area and layer regressors had values of either 0 or 1 depending on whether the dataset was for the stated area and layer, e.g. a recording from layer 5 of V1 would have regressor values (V1=1, AM=0, PM=0, MMA=0, MMP=0, RSC=0, layer=1). This explanation is now in the text as the last paragraph of the subsection “Cue-locked response amplitudes contain information about visual, motor, cognitive, and memory-related contextual task variables”.

6) It wasn't completely clear how the time of a particular cue onset was defined. In a real environment the cues would appear small (from afar) and get progressively bigger as the animal advances (at least if they are 3D objects, as depicted in Figure 1). What would be the cue onset in that case, and does the virtual environment work in the same way? This is probably not a serious issue, but it comes across as a bit at odds with the supposed "pulsatile" nature of the sensory stream, and would seem somewhat different from the auditory case with clicks.

We indeed neglected to provide this information while introducing the task, and have added this now as a third paragraph in the Results, as well as details on the following points in the Materials and methods. In summary, the “cue onset” is defined as the instant at which the cue is made visible in the virtual reality display, which is when the mouse approaches within 10cm of a predetermined cue location.

A related question concerns multiple references to cue timing made in the Introduction, as if such timing were very precise. This seems strange given that all time points depend on the running speed of the mice, which is surely variable. So, how exactly is cue position converted to cue time, and why is there an assumption of very low variability? Some of this detail may be in previous reports, but it would be important to make at least a brief, explicit clarification early on.

We have precise experimental control over the onset time of cues as we now explain in the Materials and methods (“Precision of behavioral cue timings”), and the cues were made to disappear from view in 200ms. However, there is some variability in how long any one cue will remain in the visual field of the mouse, which as the reviewer correctly noted, depends on how it runs down the maze. This variability is small except for one mouse that had a much higher running speed than other mice, and we have added Figure 1—figure supplement 1 to quantify these behavior-induced variations. Regardless, for the above reasons as well as complications due to neurons having limited receptive fields, we had included in the cue-locked response model small timing jitter parameters that allowed the model to flexibly account for some timing uncertainty in neural responses with regards to the assumed cue onset times. Insofar as we can think of, only the cue-locked response model depends on knowing the precise timings of cues, and the distribution of jitter parameters across the model fits for cue-locked neurons had a standard deviation of about 50ms, which is ballpark what we expected.

Reviewer #3:This manuscript describes measurements of neuronal activity in mice performing a discrimination task, and a new model that links these data to psychophysical performance. The key element of the new model is that sensory neurons are subject to gain modulations that evolve during each trial. They show that the model can produce pure sensory integration, Weber-Fechner performance, or intermediate states that nicely replicate the behavioral observations. This is an interesting and valuable contribution.My only significant comment relates to the Discussion, which should do more to make sure the reader understands how very different the sensory representation is in this study compared with the great majority of earlier related work in the primate:

We have added a last paragraph to the Discussion regarding some overall points where we believe our findings could be surprising compared to the primate work: (a) the small fractions of cue-locked neural activity even in V1; (b) choice modulations that are not lateralized by brain hemisphere; (c) the prevalence of many types of cue-locked amplitude modulations.

First, choice related signals are not systematically related to stimulus preferences (no Choice Probability). This is mentioned, but only very briefly.

To the best of our knowledge of the primate literature, choice probability (CP) values that correspond to higher firing rates in trials where the subject will make a choice *opposite* to the stimulus preference (CP < 0.5) could also have been detected as significant, albeit we have not been able to find reports of this other than in the recent re-analysis of monkey data by Zhao et al., 2020). If we guess correctly that CP < 0.5 is what the reviewer meant by “choice related signals [that] are not systematically related to stimulus preferences”, then we referred to such phenomena in the Discussion as cue-locked cells that have negative choice modulations as opposed to “no CP” (since these choice modulations were consistent across trials). In other words, rather than individual cue-locked cells having no CP, we observed that a substantial fraction of them had highly significant CP. Where our results differ from the primate work is in the statistics across the population of cue-locked cells, which had comparable fractions with positive (CP > 0.5) and negative (CP < 0.5) choice modulations, as opposed to the primate work where mostly CP > 0.5 results have been reported. We have added a conceptual Figure 5 as well as expanded upon these differences between our and previous work in the second-to-last paragraph of the Discussion. However it is possible that we have misunderstood the reviewer’s comment, in which case we ask for some more clarification.

Second, there appears to be no relationship between stimulus preference (visual field in this case) and noise correlation. Unfortunately, this emerges from the model fits, not an analysis of data. But is an important difference with profound implications for how the coding of information is organized. It really needs a discussion. It should also be supported by an analysis of correlations in the data. I know some people argue that 2 photon measures make this difficult, but if that's true then surely they can’t be used to support a model in which correlations are a key component.

We hope that we have not erroneously claimed any results about noise correlations in the paper, and would like to know where this was implied so that we can be more careful about the wording. We had in the past wished to perform direct analyses of noise correlations, but then realized that this was extremely difficult because our behavioral task has no exactly repeated trials that we could use to remove the effect of signal correlations. In particular, we did have in the task design multiple trials per session with exactly the same spatial configuration of cues, but unfortunately since the mice could run down the T-maze in different ways in each trial, we fear that differences in running speed, view angle etc. could result in signal-induced variability across these trials. Insofar as we can think of, we could try to subtract signal variance using a computational model, but then of course any results we obtain for noise correlations would be contingent on how well the model captures signal effects at a timepoint-by-timepoint level. We therefore feel that the soundness of any claims that we could try to make on noise correlations would be under question.

Revisions expected in follow-up work:For details, see comments 1-3 from reviewer 2 and comment 1 from reviewer 1.3) The prediction about the Fano Factor (FF) is problematic in a couple of ways. First, it seems to come out of the blue because Figure 5 is described before any discussion of the variability in the model is presented (except for the dice in the model schematic).And second, the FF prediction itself is verified for a very small fraction of neurons even when an unusual pValue of 0.1 is used. Furthermore, the mathematical derivation relies on an Taylor series around N_R_ ~ 0? (In most of the paper, CLT is invoked on the assumption N_R_ is "large"). Due to the lack of transparency of this prediction and the mild support, the authors could consider dropping it, at least from the main manuscript.

The FF measurement was indeed very difficult to perform because of insufficient statistics (that’s why we used a pValue threshold of 0.1 for testing effect sizes). This result is more of a consistency check in the sense that we didn’t observe a phenomenon that *contradicted* predictions of the theoretical model, but as noted neither do we have strong statistical support for predictions of the model. We will move the FF analysis to the supplement and word the text to indicate the difficulty of this measurement.

4) The first prediction in Figure 5 seems very unspecific. In particular, it would seem like any "open loop" modulation of the cue-locked response which depended on time, or on location along the track, would induce a trend like the one assessed in Figure 5C-E. It is not clear this is a prediction specific to the multiplicative-feedback model the authors are advocating for.

The reviewer is correct that it is always possible to write down models with ad hoc time- or location-dependent scaling of cue-locked responses (in fact, we included ad hoc location-dependent scaling in all models for the mouse data). However, what we wished to do with the feedback-loop model was to propose a neural circuit origin for the observed amplitude scaling trends, and also the specific prediction is that the cue-response amplitudes should depend on the accumulated number of cues, not time or location. We will explain more clearly in the text that this is one of the reasons why the Figure 5C-E trends were made using only neural responses to the last cue in a trial and only including trials where that occurred in the last third of the cue region, so that the spatial location of the cue along the track is kept as similar as possible. We should also add a supplementary figure where the time at which the last cue occurs (which is highly correlated with its location along the track given the stereotypical running patterns of mice) is similarly restricted.

5) The number of cells showing responses consistent with the model (Figure 5E) seems very small (~15% of 5-10% of cells with cue-locked responses). Could they really underlie the behavioral effects? The authors could perhaps comment on this.

We agree with the reviewer that these are important points to discuss in the upcoming paper. Regarding the small fraction of active cells with cue-locked responses, it is indeed intriguing that only a small fraction of neural activity even in V1 were cue-locked, but since each of our recordings only includes a very small piece of brain tissue, and 98% of recordings had at least 1 cue-locked cell (despite us having selected imaging locations agnostic to any neural analyses), the total amount of signal in the cortex can be large. Our finding of cue-locked cells in many posterior cortical regions as well as both layers 2/3 and 5 also implies that somehow the wiring of the brain allows for this small fraction of sensory-like information to be transmitted in a widespread manner (e.g. found in the retrosplenial cortex), and we might perhaps speculate that this need not have been the case for neural signals that are too weak to drive behavior.

On the small fraction of evidence-modulated cue-locked cells, we should discuss the statistical power of our analysis as well as neural circuit considerations brought up by reviewer 2’s main comments 2 and 3.

Our fitting of accumulator models to behavioral data favored the multiplicative feedback-loop (*fdbk*) model where the feedback-loop gain u*w has mean close to zero in the *fdbk* model fits. This behavioral prediction is compatible with the amplitude-vs.-cue-counts slopes of cue-locked cells (dA/dN) having a distribution with more cells having slopes closer to zero (albeit this is not necessary to generate small u*w). Given limited, noisy data we can only have the statistical power to find as significant those slopes that have large magnitudes, which can be why we only found 18% of cells with significant slopes. We should also note that this analysis was performed separately using trials of a fixed choice, i.e. testing for a dependence on cue-counts beyond that which can be accounted for by choice. However, noisy cells that receive weak accumulator feedback can more easily pass statistical tests for being modulated by choice (assuming that the accumulator drives and is therefore correlated with choice), than having count modulation beyond that explainable by choice. In these ways, we believe that our neural observations are consistent with, albeit not a necessary implication of, the behavioral model fits.

We also note that small (dA/dN) does not necessarily correspond to a small neural signal. Because (dA/dN) is a change in cue-response amplitudes per accumulated cue, if we consider the net change after accumulating ~ 8 cues (the average number of majority-side cues in the behavioral task), the responses of count-modulated cue-locked cells can increase/decrease in amplitudes by about × 2 compared to their responses to the first cue. As further discussed in the reply to reviewer 2, a feedback-loop neural circuit with very large magnitudes of dA/dN can have runaway excitation or complete suppression of cue responses after accumulating many cues. It is therefore our thinking that small dA/dN are more physiologically reasonable, and at least according to our accumulator circuit modeling results, can have a behavioral effect.

Reviewer #1:[…] 1) Figure 6 and the accompanying section of the manuscript investigate a variety of models with different architectures (feedback vs. purely feedforward) and noise sources. Here, if I understood correctly, the actual cue-driven responses are substituted with variables that are affected by different types of noise. It is this part that I found a bit disconnected from the rest, and somewhat confusing.Here, there's a jump from the actual cells to model responses. I think this needs an earlier and more explicit introduction. It is clear what the objective of the modeling effort is; what's unclear are the elements that initially go into it. This is partly because the section jumps off with a discussion about accumulator noise, but the modeling involves many more assumptions (i.e., simplifications about the inputs to the accumulators).What I wondered here was, what happened to all the variance that was carefully peeled away from the cue driven responses in the earlier part of the manuscript? Were the dependencies on running speed, viewing angle, contra versus ipsi sensitivity, etc still in play, or were the modeled cue-driven responses considering just the sensory noise from the impulse responses? I apologize if I missed this. I guess the broader question is how exactly the noise sources in the model relate to all the dependencies of the cue cells exposed in the earlier analyses.Overall, my general impression is that this section requires more unpacking; perhaps it should become an independent report.

We think that the suggested splitting off of the accumulator modeling work to a second paper is an excellent way to more cleanly separate the more complicated neurophysiological findings from the simplifications that we made in the accumulator modeling work for reasons of conceptual clarity. The modeling paper can therefore start out with an explicit list of assumptions made, as follows.

There were three major simplifications made in going from the experimentally observed cue-locked neural responses to the computational accumulator model. First, we assumed that the sensory units in the computational accumulator models only responded to one laterality of cues, because in the neural data there was only < 5 % of cells that responded to both lateralities, and even for these 5% of cells the responses still had a strong cue laterality preference. Second, while the cue-locked neurons had impulse responses of duration ~ 100ms to the pulsatile visual cues, in the computational model we simplified the sensory inputs to the accumulators to have instantaneous responses to the visual cues. The motivation for this was to make the model analytically solvable, which then allowed us to mathematically understand its phenomenology. Third, we did not separately model the other sources of cue-locked response variabilities mentioned by the reviewer, but this is because they act as behavioral sources of sensory and/or accumulator-level noise, and were thus conceptually lumped into the two (sensory and accumulator) noise sources in the models. For example, variability that is due to the mouse viewing different cues at different running speeds can be thought of as adding a different random number to a sensory unit’s response to each cue, i.e. exactly what we defined as the per-pulse sensory noise in the model. In general, sources of noise that are fast (changing from cue to cue) and have no systematic relationship to the running tally of cue counts or choice would all contribute some part of the models’ sensory noise variance. Other sources of noise that are slow (changing from trial to trial, e.g. variability that is correlated to the mouse’s eventual choice) would contribute to the models’ accumulator noise variance because they affect every cue response within the trial in the same way and therefore are accumulated in the same way as the cue responses.

Reviewer #2:[…] 1) Non-integrable variability. In addition to 'sensory noise' (independent variability in the magnitude of each pulse), it is critical in the model to include a source of variability whose impact does not decay through temporal averaging (to recover Weber-Fechner asymptotically for large N). This is achieved in the model by positing trial-to-trial variability (but not within-trial) in the dot product of the feedforward (w) and feedback (u) directions. But the way this is done seems to me problematic:The authors model variability in w*u as LogNormal (subsection “Sources of noise in various accumulator architectures”). First, the justification for this choice is incorrect as far as I can tell. The authors write: "We model m^R with a lognormal distribution, which is the limiting case of a product of many positive random variables". But neither is the dot product of w and u a product (it's a sum of many products), nor are the elements of this sum positive variables (the vector u has near zero mean and both positive and negative elements allowing different neurons to have opposite preferences on choice – see e.g., in the subsection “Cue-locked amplitude modulations motivate a multiplicative feedback-loop circuit model” where it is stated that u_i_<0 for some cells), nor would it have a LogNormal distribution even if the elements of the sum were indeed positive. Without further assumptions, the dot product w*u will have a normal distribution with mean and variance dependent on the (chosen) statistics of u and w.Two conditions seem to be necessary for u*w: it should have a mean positive but close to zero (if it's too large a(t) will explode), and it should have enough variability to make non-integrable noise have an impact in practice. For a normal distribution, this would imply that for approximately half of the trials, w*u would need to be negative, meaning a decaying accumulator and effectively no feedback. This does not seem like a sensible strategy that the brain would use.The authors should clarify how this LogNormality is justified and whether it is a critical modelling choice (as an aside, although LogNormality in u*w allows non-negativity, low mean and large variability, the fact that it has very long tails sometimes leads to instability in the values of a(t)).

We agree with the reviewer that the description of the lognormal distribution for u*w is confusing. Specifically, we wrote “limiting case of a product of many positive random variables” only as a statement about the lognormal distribution, and did not think to explain why we made that modeling choice. Upon hindsight, this way of writing it was unintentionally misleading. The justification that we had in mind when designing the model was related to what the reviewer mentioned: if the distribution of u*w can have both negative and positive tails, either the variance of this distribution must be very small relative to its (positive) mean, or else there will be a substantial fraction of trials in which there is negative accumulator feedback modulating the sensory unit responses. However, as reasoned by the reviewer, our assumption of strictly positive u*w was not very natural and would seem to require some kind of careful rectification by neural circuits, for which we have no proposed mechanism.

To address this and the other related comments below, we have extended our work to include a more thorough exploration of noise distribution modeling options, specifically the choices of (1) the sensory response distribution, which we had previously assumed to be gaussian; and (2) the feedback/modulatory noise distribution. As a reminder, the feedforward accumulator models had an accumulator state equal to n*m, where *n* is a stochastic sensory response drawn from (1) and which depends on the true stimulus counts N, whereas m is a per-trial modulatory noise drawn from (2). The feedback accumulator state is instead proportional to [exp(n m) – 1]/m, with *n* as for the feedforward models and m = u*w now interpreted as a feedback-related source of noise. We sketch our results so far below.

Our previous choice of *n* drawn from a gaussian distribution with mean proportional to *N* means that under some conditions it is possible for the sensory response *n* to be negative, which we can interpret as a nonzero probability for cues of one laterality to be confused for cues of the opposite laterality. However, this is a modeling assumption that can be tested by alternatively drawing *n* from a Gamma distribution, and interestingly seems to produce a better fit for the rat data specifically for trials where there are only cues of one laterality.

For the feedback/modulatory noise distribution (2), we tried a spectrum of options ranging from the symmetric gaussian distribution to skewed distributions with progressively longer tails, including the lognormal but also other distributions that have both negative and positive tails. Our preliminary findings are that gaussian-distributed modulatory noise actually produces a significantly better behavioral prediction for the mouse data than our previous choice of the lognormal distribution, and gaussian-distributed u*w may be more compatible with our neural observations of comparable proportions of cue-locked cells with positive vs. negative count-modulations. Specifically to answer the reviewer’s question about which are critical aspects of the lognormal distribution that produced a good fit for the rat data, by comparing its behavioral prediction to a similar model where the distribution of u*w had a truncated positive tail as well as a (smaller) negative tail, we found that these models both predicted the behavior equivalently well. This points to that the strict positivity and extreme tails of the lognormal distribution were not necessary to explain behavior, but rather other features such as a mode close to zero and a positively skewed tail were important.

2) Related to this point, it would be helpful to have more clarity on exactly what is being assumed about the feedback vector u. The neural data suggests u has close to zero mean (across neurons). At the same time, it is posited that u varies across trials ("accumulator feedback is noisy") is and that this variability is significant and important (previous comment). However, it would seem like neurons keep their choice preference across trials, meaning the trial to trial variability in each element of u has to be smaller than the mean. The authors only describe variability in u*w (LogNormal), but, in addition to the issues just mentioned about this choice, what implications does this have for the variability in u? The logic of the approach would greatly increase if the authors made assumptions about the statistics of u consistent with the neural data, and then derived the statistics of u*w.

We first admit that the result mentioned by the reviewer of cells having consistent choice preferences across trials, is unfortunately not sufficient to show that the postulated feedback strength u is fairly consistent across trials. This is because the choice- and count-modulation models that we constructed for cue-locked cell amplitudes *assumed*  across-trial consistency, e.g. cells that had differently signed choice modulations from one trial to the next would not have passed significance tests for being choice modulated. Insofar as we can think of, the only way for us to measure the distribution of the u across trials directly from the neural data, is to fit for a potentially different value of u per trial using the responses of a cue-locked cell to multiple cue presentations within a given trial. When we attempted this, say using a simple linear model *a_k_*= *a*_0_ + *uk* where *a_k_* is a given cell’s response amplitude to the *k^th^* cue in a trial, we roughly found three categories of cells with u having symmetric, positively skewed, and negatively skewed distributions, all categories with means close to zero but large variances relative to the mean. We unfortunately feel that these results have many interpretation caveats that make them difficult to claim, e.g. the many other modulatory factors that influence cue-locked activities, and independent cue-to-cue variability (intrinsic to the sensory response) that can increase the variance of the estimated u but seems very difficult to dissociate from this method of estimating u.

We therefore think that the best that we can do to answer this question is to try various modeling options for the u*w distribution as outlined in the answer to comment 1. The general idea is for the different distribution options to explore different properties such as strict positivity, long tails and so forth, using which we can determine which regimes of u*w distribution shapes produce good fits for the behavior, and then more speculatively discuss these fits with regards to neural circuit considerations as suggested by the reviewer in comment 3.

3) Overall, it seems like there is an intrinsically hard problem to be solved here, which is not acknowledged: how to obtain large variability in the effective gain of a feedback loop while at the same time keeping the gain "sufficiently restricted", i.e., neither too large and positive (runaway excitation) nor negative (counts are forgotten). While the authors avoid worrying about model parameters by fitting their values from data (with the caveats discussed above), their case would become much stronger if they studied the phenomenology of the model itself, exposing clearly the computational challenges faced and whether robust solutions to these problems exist.

We will include in the second paper a study of the phenomenology of the model in terms of different sensory response and feedback/modulatory noise distributions, as mentioned in the reply to comment 1. Our preliminary findings are that there are many combinations of distribution shapes that can produce comparably good predictions of the behavior, which may perhaps hint at robustness in the sense that the behavioral prediction depends on gross properties and not details of various noise distributions, and that multiple hypothesized neural implementations can produce the same behavioral outcomes.